

# Microphysical Parameter Choices Modulate Ice Content and Relative Humidity in the Outflow of a Warm Conveyor Belt

Cornelis Schwenk [1], Annette Miltenberger [1], and Annika Oertel [2]

[1]Institute for Atmospheric Physics, Johannes Gutenberg University Mainz, Mainz, 55099, Germany
[2]Institute for Meteorology and Climate Research Troposphere Research (IMKTRO), Karlsruhe Institute of Technology, Karlsruhe, 76131, Germany

**Correspondence:** Cornelis Schwenk (c.schwenk@uni-mainz.de)

**Abstract.** Warm conveyor belts (WCBs) play a crucial role in Earth's climate by transporting water vapor and hydrometeors into the upper troposphere/lower stratosphere (UTLS), where they influence radiative forcing. However, a major source of uncertainty in numerical weather prediction (NWP) models and climate projections stems from the parameterization of microphysical processes and their impact on cloud radiative properties as well as the vertical re-distribution of water. In this study,

we use Lagrangian data from a perturbed parameter ensemble (PPE) of a WCB case study to investigate how variations in microphysical parameterizations influence water transport into the UTLS and the outflow cirrus properties. We find that the thermodynamic conditions (pressure, temperature, specific humidity) at the end of the WCB ascent show little sensitivity to the explored parameter perturbations. In contrast, ice content and relative humidity exhibit substantial variability, primarily driven by the capacitance of ice (CAP) and the scaling of ice formation processes directly influenced by ice-nucleating particle (INP)

concentrations. Different combinations of CAP and INP scaling yield vastly different ice and relative humidity distributions at the end of the ascent and in the subsequent hours. These differences are particularly pronounced in fast-ascending air parcels, where modifications to the saturation adjustment scheme (SAT) introduce small variations in pressure and temperature at the end of ascent. Our findings have potential implications for parameter choices in cloud models and considerations for geoengineering strategies. Future comparisons with high-quality observational data could help constrain the most realistic parameter

choices, ultimately improving weather and climate forecasts.

## 1 Introduction

The most dominant greenhouse gas is water vapour (Schneider et al., 2010), and in the upper troposphere/lower stratosphere (UTLS) region, changes in its concentration lead to the most important positive feedback mechanism for climate change (Li et al., 2024; Held and Soden, 2000; Dessler et al., 2013, e.g.). Even small changes in UTLS water vapour content can signifi-

cantly alter the Earth's radiative budget (Wang et al., 2001; Hansen et al., 1984) and impact the mean and regional circulation patterns (Charlesworth et al., 2023; Ploeger et al., 2024, e.g.). It is therefore crucial to quantify i) the amount and regional distribution of moisture in the UTLS (and how this has changed over time), ii) the contributions of different transport pathways delivering moisture to the UTLS, and (iii) how these factors are likely to change in the future.



Although UTLS water vapor measurements have increased over the past 20 years (Zahn et al., 2014; Jeffery et al., 2022; Hurst et al., 2011; Tilmes et al., 2010; Konjari et al., 2024), the UTLS remains poorly characterized due to challenging measurement conditions (Jeffery et al., 2022). In situ measurements from aircraft and balloon-borne instruments provide high vertical resolution data on water vapor, but their limited regional and temporal coverage makes performing extensive climatologies difficult (Kunz et al., 2008; Zahn et al., 2014; Jeffery et al., 2022; Hurst et al., 2011). Satellite instruments offer long-term, global

datasets, but they face challenges such as coarse vertical resolution and path length limitations (Hegglin et al., 2008, 2021). Therefore, accurate measurement and quantification of UTLS moisture, as well as changes over time, and quantification of its role in climate, remains a pressing and difficult challenge.

       Moisture is transported to the UTLS through several pathways, with deep convection in the tropics being a major contributor.

Both experimental (Corti et al., 2008; Danielsen, 1993; Lee et al., 2019; Gordon et al., 2024) and modeling studies (Ueyama et al., 2018, 2023; Dauhut et al., 2018; Hassim and Lane, 2010) have shown that UTLS moistening by convection occurs not only through the direct transport of water vapor but also via the sublimation of ice crystals detrained from convective clouds. Climate models predict that UTLS humidity will increase as the climate warms due to the warming of the tropopause (Gettelman et al., 2010), but Dessler et al. (2016) also determined that a large part of this can be attributed to the increase in

sublimation of convective ice. It remains unclear how convective transport of moisture to the UTLS will change in the future, but a historical climatology by Jeske and Tost (2025) found that transport directly into the tropopause region has increased on average from 2011 to 2020 compared to the 1980s. While most of the aforementioned studies provide valuable insights into UTLS moisture transport and changes thereof, they focus primarily on the tropics. Some studies, such as Homeyer et al. (2024), who studied stratospheric hydration processes using observations over the continental United States, as well as Homeyer et al.

(2014) and Homeyer (2015), who conducted model analyses of overshooting convection in the same region, address the extratropics. However, these studies offer only a partial picture, and moisture transport in the extratropics remains insufficiently explored and measured. Addressing this gap, Zahn et al. (2014) conducted a multi-year analysis of monthly UTLS water vapor measurements and identified warm conveyor belts (WCBs) as a key contributor to extratropical UTLS moisture.

Warm conveyor belts (WCBs) are large-scale, coherently ascending airstreams that develop near extratropical cyclones (ETCs) and generate the characteristic elongated cloud bands associated with ETCs (Madonna et al., 2014). Over approximately two days WCBs transport moist air from the boundary layer poleward and upward into the upper troposphere, while the air mass undergoes complex microphysical and dynamical processes. These processes contribute to precipitation formation, influence upper-level wave propagation, and inject water vapor and hydrometeors into the UTLS. As a result, WCBs play a crucial role

by shaping mid-latitude weather systems, modulating storm development, and affecting Earth's radiative budget through cloud formation and moisture transport (Madonna et al., 2014). The ascending air produces significant amounts of precipitation, which can be hazardous (Pfahl et al., 2014); notably, WCBs account for more than 80% of the total precipitation in Northern Hemispheric storm tracks (Pfahl et al., 2014; Eckhardt et al., 2004; Binder et al., 2016). In addition to producing precipitation, WCBs transport considerable amounts of energy and various atmospheric constituents across latitudes, and influence large-

scale weather patterns (Joos et al., 2023; Madonna et al., 2014; Rodwell et al., 2018). The diabatic heating experienced by





the ascending air masses affects cyclone strength and lifetime (Binder et al., 2016; Rossa et al., 2000; Binder et al., 2023) and produces potential vorticity (PV) anomalies (Oertel et al., 2020; Methven, 2014) that could influence atmospheric blocking (Pfahl et al., 2015; Wandel et al., 2024) and Rossby wave evolution (Pickl et al., 2023; Grams et al., 2011; Joos and Wernli, 2011; Wernli, 1997). It is therefore not surprising that the incorrect representation of WCBs has been identified as a key factor in amplifying forecast uncertainties (Pickl et al., 2023; Berman and Torn, 2019; Rodwell et al., 2018; Grams et al., 2018), especially when considering cyclone evolution and heat waves (Oertel et al., 2023; Grams et al., 2018).

WCBs influence Earth's radiative budget in multiple ways. During the early stages of ascent, when WCBs are located farther south, low-level clouds predominantly exert a cooling effect. As the WCB progresses northward, high-level frozen clouds have a warming or cooling effect depending on solar insolation and cirrus optical thickness (Krämer et al., 2020; Joos, 2019). In between, mixed-phase clouds have an uncertain net radiative effect. The outflow region of a WCB is also often accompanied by ice-supersaturated regions (Spichtinger et al., 2005) and an up to 3 km deep cirrus cloud shield (Binder et al., 2020), that has a warming effect. Consequently, determining which overall effect dominates and how this balance may shift in the future remains a challenge (Joos, 2019). Additionally, it remains unquantified how much water vapor WCBs transport into the UTLS, and how this will change in the future. For these reasons, accurately representing WCBs in NWP models is crucial for both weather and climate predictions.

However, accurately representing WCBs is challenging, partly because the ascending air undergoes a wide range of warm-phase, mixed-phase, and cold-phase microphysical processes (Binder et al., 2020; Forbes and Clark, 2003; Gehring et al., 2020), producing clouds with varying phases that add significant complexity to the model representation (Oertel et al., 2023; Schwenk and Miltenberger, 2024; Hieronymus et al., 2025). Additionally, the ascent occurs across different time-scales, which result in different dominating microphysical processes (Schwenk and Miltenberger, 2024). The precise composition of WCB clouds at different stages of the ascent is therefore difficult to determine and to represent in NWP models, and hence the parameterization of the microphysical processes within them are a major source of uncertainty (Morrison et al., 2020; van Lier-Walqui et al., 2012; Posselt and Vukicevic, 2010; Oertel et al., 2025). Sensitivity experiments have demonstrated that the choice of microphysical parameterization schemes can significantly affect the characteristics of WCB ascent (Mazoyer et al., 2021, 2023), and that the diabatic processes occurring during WCB ascent are also sensitive to the specific choice of cloud microphysical parameters in a given parameterization scheme (Hieronymus et al., 2022; Neuhauser et al., 2023; Forbes and Clark, 2003). Classical sensitivity experiments, such as Monte Carlo simulations, could theoretically assess how specific parameterized processes contribute to uncertainties. However, applying these methods to NWP models across a large free parameter space, as typical for e.g. cloud microphysical parameterisations, is impractical due to their high computational cost and the vast amounts of data they produce. This has lead to the employment of Perturbed Parameter Ensembles in atmospheric sciences (PPEs e.g., Lee et al. (2011); Collins et al. (2010); Johnson et al. (2015); Oertel et al. (2025), which are sensitivity experiments that sample the multi-parameter phase space spanned by the uncertain parameter values in a statistically optimal manner with a relatively small number of simulations.





In this paper we make use of a PPE for a WCB case produced by Oertel et al. (2025), in which four cloud microphysical parameters and one parameter related to WCB inflow properties were modified. Oertel et al. (2025) specifically investigated i) how the uncertainties in microphysical parameters influence the representation of WCB ascent and ii) how the perturbed parameters affect the precipitation structure and larger-scale flow evolution. Our goal is to determine how microphysical parameter choices affect the transport of moisture and hydrometeors by ascending WCB air parcels into the UTLS. Schwenk and Miltenberger (2024) found that in the outflow of a WCB, the specific humidity ($q_v$) of Lagrangian WCB air parcels is predominantly constrained by the thermodynamic conditions at the end of ascent (i.e, the temperature and pressure in alignment with the Clausius-Clapeyron relationship). Remarkably, these results suggest that WCB air parcels are very efficient at converting water vapor into hydrometeors, otherwise one would find a larger abundance of supersaturation. This observation raises the question if this efficiency in vapor conversion reflects a physical reality, or if it is an artifact of the NWP model's potentially inadequate representation of vapor conversion in ascending air parcels?

One potentially inadequate representation could be the ICON model's saturation adjustment scheme, which forces 100% saturation over water at every model time step as long as the parcel contains any liquid water (by either evaporating cloud condensate or condensing excess vapor). However, in reality supersaturation over water can form under conditions of high vertical velocity. Making the saturation adjustment scheme more realistic by allowing for some supersaturation over water when vertical velocities are high might change the vapor conditions in the outflow of a WCB.

However, most WCB air parcels glaciate completely during their ascent, meaning that they spend a lot of time in a purely ice phase (Schwenk and Miltenberger, 2024). In this case other parameters — such as the capacitance of ice (CAP) or CCN as well as INP concentrations — likely play a more significant role in determining the efficiency of vapor conversion. For instance, an increase in the capacitance could accelerate ice particle growth, while higher CCN concentrations may result in more cloud droplets, potentially enhancing vapor conversion. Similarly, a greater concentration of INP might result in a faster transition to the ice phase, further affecting the overall conversion process. If the perturbations can significantly influence the ice content, ice number concentration or cirrus lifetime in the WCB outflow, they could also alter the radiative budget of the WCB, which would be important to consider when analyzing the future impact of WCBs on global warming. Such impacts would also be relevant for potential geoengineering strategies, such as marine cloud brightening (Latham et al., 2012) or cirrus cloud thinning (Lohmann and Gasparini, 2017; Villanueva et al., 2022; Gasparini et al., 2020), which aim to modify CCN or INP levels to change the radiative properties of clouds. The findings obtained from this paper might provide valuable insights for future geoengineering efforts seeking to manipulate cloud properties to achieve climate-modulating effects.

An additional consideration for moisture transport in WCBs is the role of convectively ascending air parcels. Schwenk and Miltenberger (2024) compared the transport of water for fast (convective) and slow ascending WCB air parcels and found that rapidly ascending WCB air parcels carry substantial amounts of ice into the UTLS. This could have notable implications for the





radiative budget of WCBs, since high level clouds can contribute to global warming (Haslehner et al., 2024), and–as discussed above–sublimating ice could increase the UTLS vapor content (Dessler et al., 2016). This raises two additional questions: i) how do different parameter choices affect the mass and number of ice particles (or snow) transported into the UTLS differently for fast and for slow ascending air parcels, and ii) how does moisture transport change when we change the fraction of convectively ascending air parcels (by modulating the SST)?

In summary, this article addresses the following research questions by analyzing a PPE of a WCB case: (How) do perturbations in microphysical parameters and SST (i) influence moisture conditions in the WCB outflow, (ii) alter hydrometeor distributions, particularly for ice, in the WCB outflow, and (iii) how do these effects vary with ascent velocity and (iv) evolve in the hours following the end of ascent?

## 2 Methods

The WCB trajectory data we investigate in this study are taken from a perturbed parameter ensemble (PPE) which is described in detail in Oertel et al. (2025). In this section, we provide a brief description of the ICON model setup, the design of the PPE, and an overview of the chosen parameter perturbations (sec. 2.1, 2.2), as described by Oertel et al. (2025) in more detail. We then proceed to explain how we select WCB trajectories and how we define the beginning and end of ascent (sec. 2.3). Finally, we explain the statistical methods used to analyze the impact of parameter choices on model output (sec. 2.4).

### 2.1 ICON model Setup and Lagrangian data

A global simulation with the Icosahedral Nonhydrostatic (ICON) modelling framework (Zängl et al., 2015; version 2.6.2.2) with approximately 13 km grid spacing (R03B07 grid) was run for 72 h, initialized from the ECMWF analysis at 18 UTC on October 3, 2016, with a 120 s time step. Two nested domains with grid spacings of approximately 6.5 km (R03B08) and approximately 3.3 km (R03B09) and time steps of 60 s and 30 s, respectively, cover the WCB ascent region and interact with the global simulation via two-way feedback. All domains have 90 vertical model levels, and in the WCB outflow region (~7-12 km) the vertical grid-spacing is between 200 and 400 m. The two moment cloud microphysics scheme from Seifert and Beheng (2006) is used, which contains six prognostic hydrometeor types (cloud liquid ($q_c$), rain ($q_r$), ice ($q_i$), snow ($q_s$), graupel ($q_g$) and hail ($q_h$)). In the global domain convection is parameterised with the Tiedke-Bechtold convection scheme (Tiedtke, 1989); in the higher-resolution nests only shallow convection is parameterised. For all other parameterisations the standard schemes are used (see Oertel et al. (2025)). The described set-up is used to perform 70 simulations with varying lower boundary conditions and cloud microphysics parameters as detailed in sec. 2.2.

The online trajectory module by Miltenberger et al. (2020); Oertel et al. (2023) is used to calculate air-mass trajectories (see Schwenk and Miltenberger (2024), Oertel et al. (2025) for details). Trajectories are started from six vertical levels between 200 and 1500 m every 1 h during the simulation in a longitude and latitude region (55 °E to -15 °E, 35 °N to 55 °N) predefined by





prior offline trajectory calculations with LAGRANTO (Sprenger and Wernli, 2015). Trajectory data is stored every 1 h. This study exclusively analyzes the Lagrangian data obtained from the PPE. An overview of the synoptic situation and trajectory

data is given in Oertel et al. (2023) and Oertel et al. (2025).

## 2.2    Perturbed parameter ensemble design

The PPE designed by Oertel et al. (2025) simultaneously perturbs the values of five parameters, which are kept constant throughout the simulation. These parameters are sea surface temperature (SST), the capacitance of ice and snow (CAP), the impact of ice nucleating particle (INP) number concentration on ice formation, cloud condensation nuclei (CCN) number con-

centration, and the saturation adjustment scheme (SAT). The specific parameter combinations for individual PPE members are chosen such that the PPE optimally explores the entire phase space spanned by the selected parameters using a Latin hypercube design. In total, 70 PPE members are available. For an overview of each parameter see Oertel et al. (2025); here we only provide a brief overview and present the chosen parameter ranges in Tab. 1.

**CAP:** The capacitance of frozen hydrometeors (CAP) influences the rate at which water vapor deposits onto ice surfaces, thereby influencing the growth of ice and snow, the phase state of the water substance in an air parcel, and the associated latent heat release. While theory predicts a normalized CAP value of 0.5 for perfectly spherical particles, realistic ice and snow hydrometeors often exhibit considerably different values (Westbrook et al., 2008; Chiruta and Wang, 2003). Consequently, in the PPE, CAP for ice and snow is varied (simultaneously) by a scaling factoring ranging from 0.2 to 1, which explores a

range for CAP from 0.1 to 0.5. CAP values for graupel and hail, which are assumed to be sufficiently spherical, are not changed.

**INP:** INP concentrations are important for cloud and precipitation formation and can alter cloud radiative forcing (CRF) (Schrod et al., 2020; Burrows et al., 2022). However, there are still large gaps in knowledge about their geographical, vertical and seasonal distributions and their representation in NWP models is highly uncertain (Wex et al., 2019, 2024; Schrod et al.,

2020; Li et al., 2022). In the PPE, Oertel et al. (2025) did not directly perturb the INP concentrations that are calculated by the ICON model, but instead scaled three processes which are known to be directly influenced by INP number concentrations-immersion freezing of cloud droplets, deposition nucleation of ice, and freezing of rain drops-by a logarithmically spaced factor ranging from 0.01 to 20. This scaling is designed to emulate scaling the number of INPs. In ICON, the heterogeneous freezing parameterization includes immersion freezing of cloud droplets (Hande et al., 2015), deposition nucleation of ice

(Hande et al., 2015) and freezing of rain drops (Bigg, 1953). In this scheme, immersion freezing is active below -12 °C as long as cloud droplets are present. The number of activated INPs is a function of temperature, and the increase in ice-number concentration per time-step depends on the current and previous number of activated INPs. Deposition nucleation takes place in a temperature range from -20 °C to -53 °C and the number of activated INPs additionally depends on supersaturation with respect to ice (Lüttmer et al., 2025). For example: at -15 °C and 110% relative humidity over ice, the immersion freezing INP

concentration amounts to approximately 5 m$^{-3}$ for the unperturbed reference and varies between 1 m$^{-3}$ and $\sim 100$ m$^{-3}$. The rate of freezing of rain is temperature dependent and below -40 °C all rain drops instantly freeze. Ice-formation parameteriza-



tions in ICON that remain unchanged in the PPE are (i) the homogeneous nucleation, parameterized following Kärcher and Lohmann (2002) and Kärcher et al. (2006), where the freezing rate and critical supersaturation are functions of temperature; (ii) homogeneous freezing of cloud droplets (Jeffery and Austin, 1997), where below -50 °C, all cloud droplets freeze instantly;
and (iii) secondary ice production (Hallet and Mossop rime splintering), active between approximately -8 °C and -3 °C (Hallet and Mossop, 1974; Seifert and Beheng, 2006).

**CCN:** The cloud droplet activation scheme implemented in ICON was originally designed for continental Germany by Hande et al. (2016) and assumes a vertical CCN profile that decreases as pressure decreases. This profile does not change over time nor
across the globe, and droplet activation scales with the resolved vertical velocity. However, it is known that CCN number concentrations are not constant, but vary in space and time, depending on factors such as the season and air mass origin (Schmale et al., 2018; Rose et al., 2021). Over continents they can also differ substantially from concentrations over the open ocean. Therefore, Oertel et al. (2025) used a modified cloud droplet activation scheme designed by Oertel et al. (2023) to account for CCN concentrations over the northern Atlantic. The details are described in the latter publication, but we point out that,
in the new scheme, cloud droplet activation continues to scale with pressure and vertical velocity, and the profile remains the same globally (i.e. it is a tailored adjustment for an open-ocean WCB). In the unperturbed reference, the CCN concentration is approximately $\sim 250\,\mathrm{cm}^{-3}$ near the cloud-base (at $\sim 800\,\mathrm{hPa}$) for large vertical velocities (see Fig. 2 in Oertel et al. (2023)). In this PPE, the uncertainty in representing CCN number concentrations is accounted for by scaling the number of activated cloud droplets per time step in the entire profile by a factor ranging from 0.4 to 20. This results in CCN concentrations varying
from approximately $100\,\mathrm{cm}^{-3}$ to $5{,}000\,\mathrm{cm}^{-3}$.

**SAT:** The saturation adjustment scheme used in ICON, as discussed in the introduction, instantaneously removes super- or (in-cloud) sub-saturation by condensing or evaporating excess or deficient water vapor until a relative humidity of 100% is achieved, which becomes increasingly unrealistic when vertical velocities are high or CCN numbers very low. In the PPE, this
scheme is modified such that when vertical velocities are high some supersaturation can form. This is done by including a scaling factor $f_{SATAD}$, ranging from 0 to 0.1, which is multiplied with the vertical velocity (only when vertical velocities are larger than zero), added to one, and finally multiplied with the specific humidity from the default scheme:

$$q_{v,s}^*(T,p) = (1 + (f_{SATAD} \cdot w)) \cdot q_{v,s}(T,p)\,|\,w > 0. \tag{1}$$

A factor $f_{SATAD}$ of 0 corresponds to the standard scheme, which inhibits the formation of supersaturated conditions.

**SST:** Tropospheric humidity in the inflow region of a WCB is crucial for the subsequent ascent (Schäfler and Harnisch, 2014; Christ et al., 2025; Berman and Torn, 2022, 2019) and formation of precipitation (Dacre et al., 2019). Yet NWP models can have substantial humidity errors (Schäfler et al., 2010, 2011). Since the moisture content of marine boundary-layer air is mostly controlled by temperature, the PPE modifies SST (which in ICON is kept constant throughout the simulation) within





| Parameter | Min | Max | Unit | Process |
|-----------|-----|-----|------|---------|
| SST | -2 | +2 | K | sea surface temperature/humidity/latitude |
| CCN | 0.4 | 20 | 1 | cloud droplet activation |
| INP | 0.01 | 20 | 1 | ice nucleation |
| CAP | 0.2 | 1 | 1 | growth of ice/snow by vapour deposition |
| SAT | 0 | 0.1 | 1 | saturation adjustment scheme |

**Table 1.** Overview of parameter perturbations (taken from Oertel et al. (2025)).

the range of $\pm 2\,\mathrm{K}$ to analyze the sensitivity to both the temperature and moisture in the WCB inflow region. A difference in SST can also be seen as analogous to a shift in latitude or as an uncertainty in cyclone position or WCB inflow location.

### 2.3 Trajectory selection and ascent time characterisation

WCB trajectories must fulfill the criteria of ascending at least $600\,\mathrm{hPa}$ within $48\,\mathrm{h}$. We use the widely used ascent time-scale $\tau_{600}$, defined as the shortest time taken per trajectory to ascend at least $600\,\mathrm{hPa}$, to characterise the ascent time of a WCB

trajectory. As we are most interested in the characteristics of trajectories once they have completed their ascent and have entered the UTLS, we do not only consider the $\tau_{600}$ ascent-segment, but the entire time around this segment during which the ascent velocity remains above $8\,\mathrm{hPa\,h}^{-1}$ (using the algorithm described by Schwenk and Miltenberger (2024)). We refer to this time as $\tau_{\mathrm{WCB}}$. The end of ascent is defined as the last time-step in $\tau_{\mathrm{WCB}}$. Note that in contrast, Oertel et al. (2025) consider the $\tau_{600}$ segment.

### 240 2.4 Methods for quantifying parameter perturbation impacts

The impact of parameter perturbations are investigated for selected properties of WCB outflow (target variables): temperature, pressure, specific humidity, relative humidity over ice and hydrometeor content at (i) the end of WCB ascent (see 2.3), (ii) during the ascent and (iii) 5 h after the end of ascent. The distribution of these values across all selected trajectories is considered as are summary statistics characterizing the distribution. The latter are particularly useful for further quantification of the param-

eter perturbation impact, for which we use Spearman correlation coefficients and random forest regression models. For each PPE member, the mean, median, standard deviation, as well as the $5^{\mathrm{th}}$, $25^{\mathrm{th}}$, $75^{\mathrm{th}}$, and $95^{\mathrm{th}}$ percentiles over all trajectories (in that PPE member) are calculated for the target variables. Taking the specific humidity $q_v$ as an example, we use the following notation for these statistics: $q_v^{\mathrm{mean}}$, $q_v^{\mathrm{median}}$, $q_v^{\mathrm{std}}$, $q_v^{5\mathrm{th}}$, and so on. The value of the variables are usually taken at the end of ascent (e.g ice content, temperature), but some are also taken during the ascent (e.g temperature at 99% glaciation, maximum

hydrometeor content achieved during ascent) or after the ascent (e.g ice content 5 h after completing $\tau_{\mathrm{WCB}}$). If the latter is the case this is explicitly stated in the text. In certain cases, trajectories from multiple PPE members are examined in detail (as opposed to looking only at the summary statistics), particularly to investigate how distributions change with changing parameters.





**Spearman correlation**

One metric for determining the impact of a parameter on the summary statistic of some target variable is the Spearman correlation coefficient ($R_s$). In contrast to the "regular" correlation coefficient, it ensures that also non-linear but monotonic correlations (e.g., an exponentially decreasing correlation) result in a high coefficient. If a parameter choice strongly influences a variable (monotonically), then $R_s$ for the relevant summary statistic will be high. To see if the impact of the correlation is noticeable compared to the spread of values within an ensemble member, we use a scatter plot of mean values with shaded
percentiles.

**Random Forest Regression Models**

If multiple parameters impact a variable in a complex and non-linear way, using simple correlation coefficients is not sufficient. We therefore also employ random forest (RF) regression models (Breiman, 2001), which can capture complex and
non-linear interactions between the perturbed parameters and the target variables. To determine which PPE-parameters had the most influence on the RF-output, we examine the impurity-based feature importance (IBF) scores, which quantify the reduction in variance contributed by each parameter across all decision trees in the forest. Note: a high IBF importance score for a PPE-parameter does not necessarily imply a clear or strong correlation with the output variable. Instead, it indicates that the PPE-parameter contributes significantly to the RF-model's predictions, possibly by interacting with the impact of other PPE-
parameters or by affecting the RF-model's decision-making structure, and is only a meaningful metric when the RF-prediction is good. We use partial dependence plots (PDPs) to visualize interactions between the target response $X$ and the selected perturbed parameters (Friedman, 2001). To construct a PDP, the model prediction is calculated for a range of values of the feature of interest, while holding all other features constant. This process is repeated across all possible values of the chosen feature, and the average predicted response is plotted against the feature values. The resulting PDP visualizes the effect of the selected
feature on the target response, providing insights into the feature's influence and possible nonlinear relationships within the model. A two-dimensional PDP can help in identifying joint parameter impact on a target variable.

In summary, the purpose of our RF analysis is mostly to identify whether parameter changes have an impact, and if so, which parameters are the most important. We take the following approach:

– Train RF model for target variable $X$, called $\mathrm{RF}^1_X$ (for example $\mathrm{RF}^1_{\mathrm{mean(qv)}}$).

    – Define most important perturbed parameters as those with an IBF score larger than 0.1.

    – Train a new RF model using only these perturbed parameters with the same target variable ($\mathrm{RF}^2_X$).

    – Evaluate whether said parameters have a strong effect on target variable by looking at $R^2$ (linear correlation coefficient) and root mean square deviations (NRMSE) between $\mathrm{RF}^2_X$ prediction and actual values (average over 100 model
285       iterations).

    – (Optional) construct and interpret PDPs using $\mathrm{RF}^2_X$.



It is important to clarify that we are **not** using the random forest model to make predictions in the conventional sense, especially as our analysis is conducted on the training data itself. Instead, if the model prediction is adequate, we assume that the model is able to find relationships between the output variables and the perturbed parameters. By examining feature importance and partial dependence plots, we aim to investigate these patterns, rather than to predict outcomes on new data. We selected model configurations that balance RF-model complexity with the need for generalization given the small dataset size (70 data points when using summary statistics and six perturbed parameters). Each model was configured with 100 trees to ensure sufficient coverage of feature space, we set the maximum tree depth to 3 to prevent over-fitting and specified a minimum sample split of 10, which constrains each split to be based on a meaningful subset of the data.

## 3  Properties of WCB outflow at the end of ascent

In this section we provide an overview of the properties of WCB trajectories at the end of ascent, since this is where we focus most of our analysis. Spatially, the end of the ascent is spread across a large area, ranging in longitude from $60^\circ$ W to more than $45^\circ$ E (Fig. 1,c). However, the core of end-of-ascent longitude positions are focused around $60^\circ$ W to $10^\circ$ W (Fig. 1 a) with only very few extending further east. In latitude, trajectories finish their ascent between $35^\circ$ N and $85^\circ$ N (Fig. 1 c), with largest density at approximately 40, 50, 58 and $68^\circ$ N (Fig. 1 b), indicating the presence of large-scale organization in the ascent of air within the WCB. For the discussion of changes in the positions of WCB trajectories with parameter perturbations, the reader is directed to Oertel et al. (2025), since this is not the area of focus in this study.

At the end of ascent, trajectories across all PPE members ascend to pressures between approximately 200 and 400 hPa (Figure 2 c and d). Temperatures range from approximately -60 to -30 °C; the majority of trajectories (ca. 75%) have temperatures below the homogeneous freezing temperature of -38 °C (Figure 2 a and b). Trajectories are very dry when they complete their ascent, with an average $q_v^{\mathrm{mean}}$ of 0.23 g kg$^{-1}$ (Figure 2 e and f). However, the distribution of $q_v$ in each PPE member is skewed towards higher values, meaning that although the median $q_v$ at the end of ascent ranges from 0.17 to 0.20 g kg$^{-1}$ for PPE members, all PPE members also have (few) trajectories with $q_v$ larger than 0.6 g kg$^{-1}$ at end of ascent. We discuss the effects of parameter perturbations on $T$, $p$ and $q_v$ in Section 3.1.

Almost all trajectories are fully glaciated by the end of their ascent and less than 1 % of all trajectories have cloud droplet mass concentrations ($q_c$) larger than zero (not shown). Trajectories at end of ascent also contain no graupel and no rain (not shown). The only relevant hydrometeor species are therefore ice and snow, with the former being the most predominant (with ice mass mass mixing ratio $q_i^{\mathrm{mean}}$ and ice number concentration $N_i^{\mathrm{mean}}$ being on average 0.046 g kg$^{-1}$ and $2.49 \cdot 10^6$ kg$^{-1}$, compared to $q_s^{\mathrm{mean}}$ and $N_s^{\mathrm{mean}}$ which are on average 0.001 g kg$^{-1}$ and 540 kg$^{-1}$). We therefore focus our analysis on $q_i$ at the end of ascent, which shows a peak at small values close to $\sim 0.05$ g kg$^{-1}$ and a long tail to larger values of up to around 0.3 g kg$^{-1}$ for the PPE member with the highest mean $q_i$ (Figure 2 g). Hydrometeor contents are therefore roughly one order of magnitude smaller than $q_v$. In contrast to $q_v$, we find that $q_i$ varies much more strongly between PPE members (Figures 2 g





and h). The standard deviation of $q_i^{\mathrm{mean}}$ across all PPE members is 0.01 g/kg (vs. 0.004 g/kg for $q_v^{\mathrm{mean}}$) and the highest $q_i^{\mathrm{mean}}$ (0.080 g/kg) is almost 3 times larger than the smallest $q_i^{\mathrm{mean}}$ (0.027 g/kg). This corroborates the visual impression from Fig. 2 h of substantial changes in the $q_i$ distribution in particular with respect to the occurrence of high $q_i$ values. We assess the effects of parameter perturbations on the hydrometeor content at the end of ascent in Section 3.2.

The mean relative humidity over ice ($RH_i$) for all PPE members is larger than 100%, but also here we find much stronger differences between PPE members (Figure 2 i and j) than for $T$, $p$ and $q_v$, which is particularly interesting given that the differences in $q_v$ are small. The smallest $RH_i^{\mathrm{mean}}$ of 101.2% is not only much smaller than the largest (109.8%), but the two distributions are also markedly different (Figure 2 i): The $RH_i$ distribution from the PPE member with the smallest $RH_i^{\mathrm{mean}}$ is relatively narrow with only few trajectories exceeding 110 % $RH_i$, while the distribution from the PPE member with the

largest $RH_i^{\mathrm{mean}}$ is very broad featuring $RH_i$ values above 110 % (120 %) in 44 % (15 %) of the trajectories. We interpret the effects of parameter perturbations on the relative humidity at the end of ascent in Section 3.3.

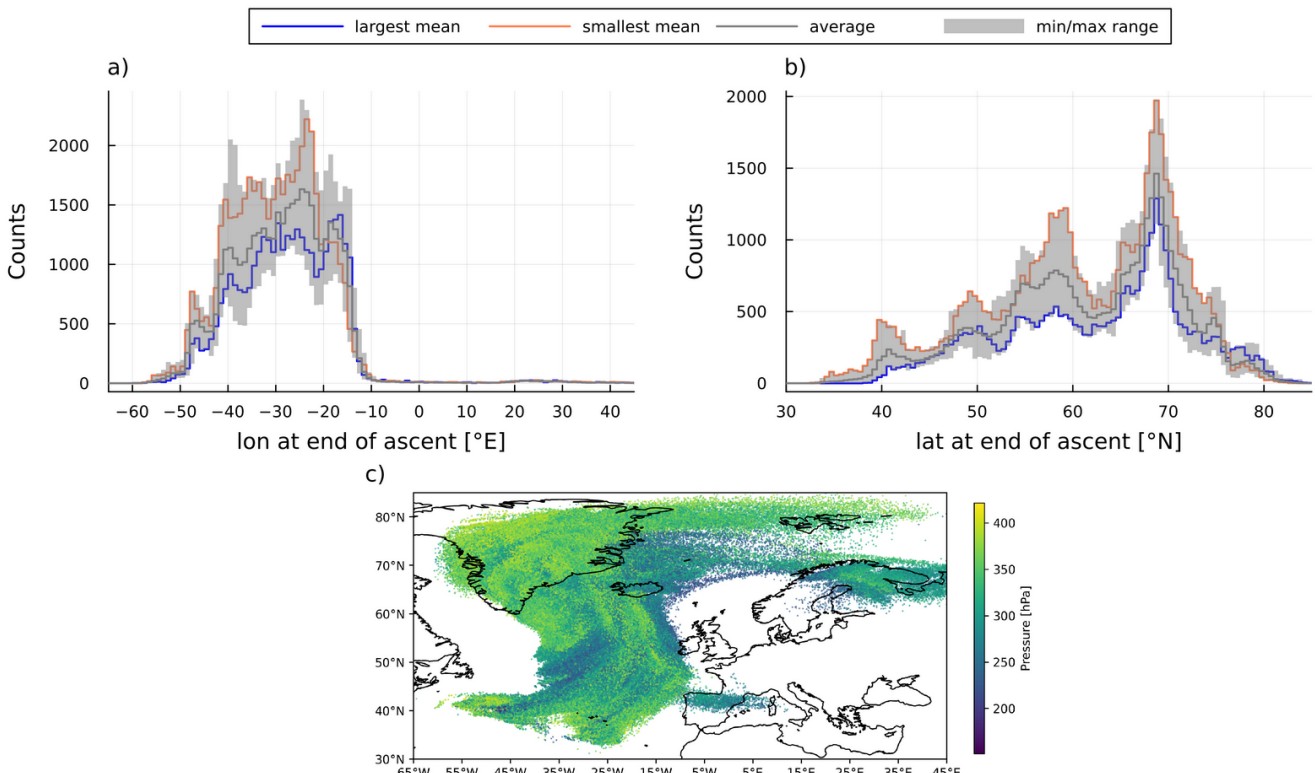

**Figure 1.** Histograms for a) longitude and b) latitude at the end of ascent, with each showing distributions for PPE members with the largest mean (blue) and smallest mean (orange), as well as the average distribution in grey and the min/max distributions per variable bin in the shaded area. c) shows the trajectory positions at the end of the ascent for all PPE members (note that trajectories finish their ascent at different times in the simulation), with pressure as the color scale.

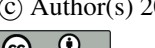


**Figure 2.** Histograms for a) temperature $T$, c) pressure $p$, e) specific humidity $q_v$, g) ice mass mixing ratio $q_i$ (note the logarithmic x-axis) and i) relative humidity over ice $RH_i$ at the end of ascent. Each figure shows distributions for PPE members with the largest mean (blue) and smallest mean (orange), as well as the average distribution in grey and the min/max distributions per variable bin in the shaded area. b), d), f), h) and j) show $T^{\text{mean}}$, $p^{\text{mean}}$, $q_v^{\text{mean}}$, $q_i^{\text{mean}}$ and $RH_i^{\text{mean}}$ per PPE member, respectively, sorted according to the mean, with $5^{\text{th}}$ to $95^{\text{th}}$ percentiles and inter-quartile range shaded in grey.





### 3.1 Parameter effects on temperature, pressure and specific humidity at the end of ascent

As discussed above, $T$, $p$ and $q_v$ show little variation between PPE members (Figure 2 b, d and f), with the variability within
each PPE member being significantly larger than the mean differences between PPE members. The distributions for PPE members with the highest and smallest mean values for $T$, $p$ and $q_v$ are also very similar (Figs. 2 a, c and e). The relatively large spread of the distribution (min/max shaded areas in Fig.2 a,c,e ) is a result of the differing number of WCB trajectories per PPE member (which is primarily controlled by SST, see Oertel et al. (2025)). Yet, the shape of the distributions remains similar across the entire PPE. We interpret this inability of the parameters to change the pressure, temperature, and vapor conditions
at the end of the WCB ascent as an indication that the thermodynamic conditions in the outflow of a WCB are largely constrained. Any larger $R_s$ values found for $T^{\mathrm{mean}}$ or $p^{\mathrm{mean}}$ versus a parameter perturbation, while mathematically valid, do not take into account the immense spread of the $5^{\mathrm{th}}$ to $95^{\mathrm{th}}$ percentiles and inter-quartile ranges (Fig. 2 b and d). The $\mathrm{RF}^2_{\mathrm{mean(p)}}$ and $\mathrm{RF}^2_{\mathrm{mean(T)}}$ model predictions (which we use to determine which parameters have the strongest influence on a variable, as long as the model prediction is strong) are also relatively weak, with mediocre $R^2$ and large NRMSE-values (Tab. 2), meaning
that also the RF model does not find meaningful changes in $T^{\mathrm{mean}}$, $p^{\mathrm{mean}}$ and $q_v^{\mathrm{mean}}$ with parameter perturbations.

However, some relationships are worth noting. For example, $T^{95\mathrm{th}}$ and $q_v^{95\mathrm{th}}$ show an unmistakable correlation with SST perturbations (Fig. 2,c) and to each other (Fig. A1 a). Oertel et al. (2025) found that increasing SST leads to higher potential temperatures and a greater number of WCB trajectories, as enhanced latent heat release facilitates more cross-isentropic ascent.
Consequently, in PPE members with higher SST, more trajectories are able to meet the 600 hPa ascent threshold that would otherwise fall short at lower SST values. We therefore attribute the increase in $T^{95\mathrm{th}}$ and $q_v^{95\mathrm{th}}$ with SST to the larger number of WCB trajectories that end their ascent at relatively high pressures ($\sim$400 hPa) and lower latitudes ($\sim$40°N) in high-SST simulations (not shown). These trajectories are more likely to be absent from the WCB trajectory set in PPE members with lower SST. The change is stronger for $q_v^{95\mathrm{th}}$ than for $T^{95\mathrm{th}}$ because of the exponential dependency of the saturation specific
humidity with temperature and the additional modulation of $q_v$ by $RH_i$ at a given temperature.

An important finding from a previous analysis of $q_v$ in WCB outflow by Schwenk and Miltenberger (2024) was that $q_v$ values are largely controlled by temperature. We find the same in the present PPE: the specific humidity for the ensemble members with the highest and lowest $q_v^{95\mathrm{th}}$ value correlates strongly with the calculated saturation specific humidity (using
parcel temperature $T$ and pressure $p$, Figure A1,b). This indicates that in both PPE members the specific humidity at the end of ascent is predominantly constrained by the thermodynamic conditions, i.e., changes in $q_v$ are a result of changes in temperature.

In summary, the parameter perturbations in our PPE only have a limited effect on changing the distributions of $T$, $p$ and $q_v$ at the end of ascent, indicating that the conditions at the end of ascent experience strong thermodynamic constraints.





| | $T^{\mathrm{mean}}$ | $p^{\mathrm{mean}}$ | $RH_i^{\mathrm{mean}}$ | $q_v^{\mathrm{mean}}$ | $q_i^{\mathrm{mean}}$ | $Ni^{\mathrm{mean}}$ |
|---|---|---|---|---|---|---|
| CAP | 0.295 | 0.143 | 0.319 | 0.115 | 0.486 | 0.434 |
| INP | 0.384 | 0.249 | 0 304 | 0.322 | 0.251 | 0.287 |
| CCN | | 0.299 | 0.229 | | 0.100 | 0.131 |
| SST | 0.136 | 0.132 | | 0.382 | | |
| SAT | 0.112 | 0.177 | | | | 0.100 |
| $R^2$ | 0.612 | 0.558 | 0.765 | 0.583 | 0.727 | 0.700 |
| NRMSE | 0.618 | 0.660 | 0.481 | 0.641 | 0.518 | 0.542 |

**Table 2.** Impurity-based feature importance (IBF) scores of CAP, INP, CCN, SST and SAT derived from $\mathrm{RF}_X^1$ for a selection of mean variables. Only scores $\geq 0.1$ are included. The bottom rows show the $R^2$ and NRMSE value for the predictions of the second iteration of the forest regression model ($\mathrm{RF}_X^2$) using only parameters with IBF $\geq 0.1$. Note: all of these values are slightly different each time a forest model regression is performed, the values shown here are averaged over 100 iterations.

## 3.2 Effect of parameter perturbations on the hydrometeor content (during and at the end of the ascent)

At the end of ascent, virtually all hydrometeors are ice (Sec. 3). The summary statistics as well as distributions of mass mixing ratios $q_i$ show large differences between PPE members (Fig. 2 g and h) suggesting that the parameter perturbations influence the hydrometeor content in the outflow of the WCB. The IBF scores (Tab. 2) indicate that CAP and INP scaling factors are the most important parameters for the mean ice number ($N_i$) and mass ($q_i$) concentration. The CCN scaling factor shows an IBF score of approximately 0.1 for these variables, indicating that it is less important in the frozen-phase than it is in the liquid-phase and mixed-phase cloud regime (as is shown later in this section).

**Impact of CAP on ice properties at end of ascent**

Increasing CAP clearly decreases $q_i^{\mathrm{median}}$ at the end of the ascent while the spread in values remains largely unaltered (Fig. 3 a, note logarithmic y-axis). The spread (in the percentiles) is unchanged because the distribution and shape of $q_i$ values remains similar for large and small CAP (Fig. 3 d). Increasing CAP also strongly decreases $N_i^{\mathrm{median}}$ at the end of the ascent and reduces the spread (Fig. 3 b), which is a results of a large difference in the distribution shape of $N_i$ values for PPE members with large and small CAP (Fig. 3 e). When CAP is small (<0.4) the distribution of $N_i$ shows a large spread and hints at bi-modality, with one peak at approximately $10^5 \, \mathrm{kg}^{-1}$ and another weak peak at approximately $5 \cdot 10^6 \, \mathrm{kg}^{-1}$. When CAP is large ($\geq 0.8$) the bi-modality almost disappears, reducing the spread. The peak of the distribution also shifts to smaller $N_i$ for PPE members with larger CAP. The radius $r_i$, on the other hand, increases clearly with CAP, as does the spread (Fig. 3 c and f).

We interpret these findings as follows: An increase in CAP enhances the deposition rate, which, in the mixed-phase, can accelerate the Wegener–Bergeron–Findeisen (WBF) process. Consequently, when CAP is high, ice crystals grow more rapidly,





depleting vapor more efficiently and accelerating the evaporation of cloud droplets. This leads to an earlier onset of full glacia-tion (see Supplement Fig. SI1) — defined as the first time step when $q_c$ reaches zero — at higher pressures and temperatures. Thus, fewer cloud droplets reach the homogeneous-freezing level of -38 °C, where they would otherwise freeze and generate numerous small ice crystals. This explains why the distribution appears slightly bi-modal at low CAP values — indicating

many small cloud droplets reach the homogeneous-freezing level — but becomes uni-modal at high CAP values. Furthermore, an increased deposition rate leads to the formation of larger ice crystals (reflected in the increase of $r_i$ with CAP), which have higher terminal velocities and precipitate more quickly. This precipitation process reduces the peak number concentration of ice crystals $N_i$ and, ultimately, the ice mass mixing ratio $q_i$.

**Impact of INP on ice properties at end of ascent**

The effects of varying the INP scaling factor exhibit similarities to those observed for CAP, but with some notable differences. As for CAP, an increase in the INP scaling factor leads to a decrease in $q_i^{\mathrm{median}}$, albeit to a lesser extent than CAP (Fig. 4 a). Similarly to CAP, low INP scaling factor values result in a distribution of $N_i$ that is slightly bi-modal with a high spread, and high values result in a uni-modal distribution with a low spread (Fig. 4 b). However, the key difference is that the peak of the

$N_i$ distribution shifts in the opposite direction, i.e. towards smaller $N_i$ values with an increase in INP scaling factor (Fig. 4 e, Fig. SI2). Although the INP scaling factor has a stronger effect on the location of the peak $N_i$ than CAP (Fig. SI2), it shows no clear relationship with $N_i^{\mathrm{median}}$ values (Fig.4 b) because the spread of values is much larger and, due to the exponential nature of $N_i$-values, this means that the peak does not correspond to the mean. The peak for $r_i$ shifts to smaller $r_i$ values with a decrease in INP scaling factor, (Fig. 4 f), but the spread is larger which is why this trend is less apparent in $r_i^{\mathrm{median}}$ (Fig. 4 c).


We interpret these findings as follows: By design, increasing the INP scaling factor enhances the immersion freezing of cloud droplets, the deposition nucleation of ice, and the freezing of raindrops (sec. 2), thereby accelerating the conversion of cloud droplets to ice. Similarly to CAP, a higher INP scaling factor leads to an earlier onset of full glaciation (see Supplement Fig. SI1) with fewer cloud droplets reaching the homogeneous-freezing level, where they would otherwise form numerous

small ice crystals. The explanation for the bi-modal distribution of $N_i$ at low INP scaling factor values and the transition to a uni-modal distribution at higher values is therefore the same as for CAP. The only addition is that a smaller INP scaling factor shifts the location of the peak $N_i$ to smaller values (a smaller CAP shifts it to larger values) and also exerts a stronger control on it (Fig. SI2). If this peak is interpreted as those trajectories that encounter no (or less) homogeneous freezing of cloud droplets, then its location should primarily increase with INP scaling, which is what we observe. Furthermore, since hydrometeor growth

by deposition is more efficient than by condensation, and a higher INP scaling factor prolongs the duration spent in the ice phase, it ultimately promotes the formation of larger ice crystals, contributing to the observed increase in $r_i$ and the slight decrease in $q_i$ (heavier ice particles fall out more quickly).







**Figure 3.** a) $q_i$, b) $N_i$ and c) ice radius ($r_i$), over CAP, with makers showing median values and shaded areas the $5^{th}$ to $95^{th}$ percentiles (light grey) and $25^{th}$ to $75^{th}$ percentiles (dark grey). d) e) and f) show $q_i$, $N_i$ and $r_i$ respectively, but as histograms for all PPE members with CAP values either larger than 0.8 (red; in total 16 PPE members and 633,400 trajectories) or smaller than 0.4 (cyan; in total 18 PPE members and 696,846 trajectories).



**Figure 4.** a) median $q_i$, b) median $N_i$ and c) median ice radius ($r_i$), over CAP. Makers show median values, shaded areas show $5^{\text{th}}$ to $95^{\text{th}}$ percentiles (light grey) and $25^{\text{th}}$ to $75^{\text{th}}$ percentiles (dark grey). d) e) and f) show $q_i$, $N_i$ and $r_i$ respectively, but as histograms for all PPE members with INP scaling factors either larger than 1 (red; in total 29 PPE members and 1,123,557 trajectories) or smaller than 0.5 (cyan; in total 35 PPE members and 1,417,123 trajectories).



**Impact of CCN on ice properties during the ascent**

The third most important parameter (based on the IBF scores, Tab. 2) for the ice content at the end of the ascent is CCN. How-
ever, the CCN scaling factor primarily influences the hydrometeor content and mixed-phase microphysics during the WCB
ascent as opposed to at the end. It correlates almost perfectly with the mean of maximum number concentration of hydromete-
ors achieved during the ascent ($R_s = 0.9994$, Fig, 5 d). This clear dependency comes from the fact that an increase in the CCN
scaling factor primarily increases the number of cloud droplets, while not strongly affecting the liquid water content in early
stages of the parcel evolution (Fig. 5 e). Therefore, cloud droplets are far smaller, which delays the formation of rain and en-
hances the residence time of liquid condensate. This is also the reason why air parcels begin their glaciation (defined as first time
when ($q_c + q_r/q_{\text{tot}}$) < 0.5) at lower pressures and temperatures (i.e., later in the ascent) when the CCN scaling factor is larger
(Fig. 5 a and b). This in turn leads to larger mean maximum mixing ratios of supercooled liquid water (max(SLW)) achieved
during the ascent with an increasing CCN scaling factor (Fig. 5 c). The mean maximum mixing ratio for snow (max($q_s$)) during
ascent also increases with CCN (Fig. 5 e), presumably because in the two-moment microphysics scheme riming contributes to
the mass growth rate of snow, which presumably happens more often when the CCN scaling factor is high due to the larger
abundance of cloud droplets. However, this large effect of the CCN scaling factor on the hydrometeor population is mostly lost
by the end of the ascent, showing that once air parcels glaciate, CAP and INP dominate the ice-phase cloud microphysics.

**(Non-)Impact of SST on ice properties**

Another important result from the analysis of parameter effects on the hydrometeor population is that changes to SST, which
substantially increase the amount of WCB trajectories that ascend quickly (Oertel et al., 2025), show no effect on distributions
of $N_i$ and $q_i$ at the end of ascent. This is interesting because Schwenk and Miltenberger (2024) found that fast ascending
trajectories ($\tau_{600} < 5\,\text{h}$) have larger hydrometeor content at the end of the ascent than more slowly ascending trajectories
($\tau_{600} > 20\,\text{h}$). Since Oertel et al. (2025) show a shift towards shorter ascent timescales and enhanced convective activity for
increasing SST, an impact on $N_i$ and $q_i$ at end of ascent would have been expected between PPE members with low and high
SST values. However, the present study indicates that the impact of CAP and INP dominates. However, we note that for the
studied WCB only 5% of trajectories have an ascent timescale $\tau_{600}$ below 10 h. This fraction only changes slightly for PPE
members with an SST perturbation below -1°C (4.5%) and larger than 1°C (6.4%). We hypothesize that an effect of SST on $N_i$
and $q_i$ could be present for a more convectively active WCB case. We further emphasize that changes in SST have no apparent
effect on the temperature and pressure at which air parcels glaciate (see Supplement Fig. SI7).

### 3.3 Effect of parameter perturbations on relative humidity over ice ($RH_i$) at end of ascent

Despite the strong temperature control on $q_v$ values (Sec. 3.1), previous work suggests that substantial supersaturation over ice
(up to 120 %) can exist in WCB outflow (Schwenk and Miltenberger, 2024; Spichtinger et al., 2005). This is also evident in the
PPE, where we find large $RH_i$ values as well as a strong variability in $RH_i$ values and $RH_i$ distributions between PPE mem-
bers (Figure 2 i and j). The parameters with the largest IBF scores (Tab. 2) for mean $RH_i$ are CAP (0.319), INP (0.304) and
CCN (0.229), and the $\text{RF}^2_{\text{mean}(RHi)}$ model has a good predictive capability, with $R^2 = 0.765$ and NRMSE = 0.481. The PDPs



**Figure 5.** Scatter plots of a) mean pressure at 50 % glaciation $((q_i + q_s + q_g)/q_{tot} = 0.5)$, b) mean temperature at 50 % glaciation, c) mean of maximum total hydrometeor mass mixing ratio during the ascent and d) mean of maximum total hydrometeor number concentration during the ascent over CCN scaling factors. Shaded areas show $5^{th}$ to $95^{th}$ percentiles (light grey) and $25^{th}$ to $75^{th}$ percentiles (dark grey). Spearman correlation coefficients for CCN scaling factors with individual mean maximum mass mixing ratios during the ascent for all hydrometeors are shown in (e)





show that increases in CAP, INP and CCN all work to decrease $RH_i$ (see Supplement Fig. SI3). The two-dimensional PDPs (Fig. 6) additionally indicate that these parameters have a joint impact: an increase in CAP, INP and CCN is most effective in reducing $RH_i^{\text{mean}}$ when the respective other two parameters are also increasing (lowest $RH_i^{\text{mean}}$ values seen only in top-right corners of Fig. 6).

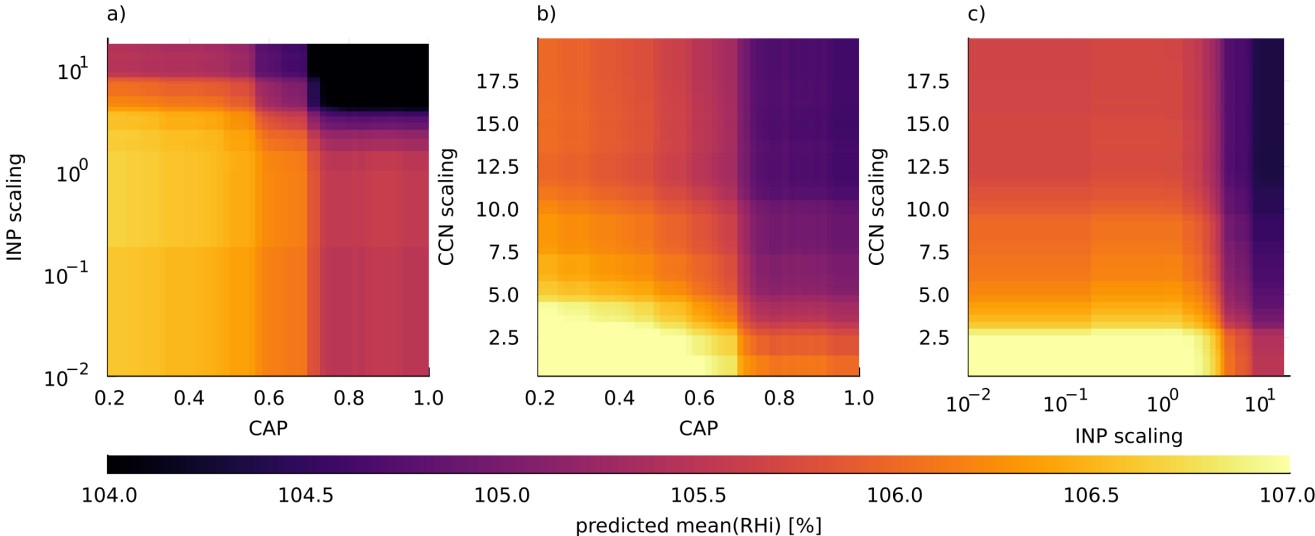

**Figure 6.** Two dimensional PDPs for $\text{RF}^2_{\text{mean(RHi)}}$ for INP vs CAP (a), CCN vs CAP (b) and CCN vs INP (c).

However, when considering the median $RH_i$ values instead of the mean, the joint impacts of parameter perturbations turn out to be more nuanced. An increase in CAP values only reduces the median $RH_i$ when the INP scaling factor is larger than one (red dots in Fig. 7 a). When the INP scaling factor is smaller than one, an increase in CAP slightly increases the median $RH_i$ (cyan dots in Fig. 7 a). The same is true the other way around: an increase in INP scaling factor increases the median $RH_i$ when CAP is smaller than 0.4 (cyan dots in Fig. 7 b) and only shows a decrease in median $RH_i$ when CAP is larger than 0.4 (red dots in Fig. 7 b). We also find that the spread of $RH_i$ values reduces strongly with an increase in CAP and INP (shaded areas in Figs. 7 a and b). This behaviour is not seen for the CCN scaling factor, which reduces the median $RH_i$ regardless of CAP and INP scaling and does not influence the spread (Fig. 7 c). The fact that the mean, median and spread of $RH_i$ values are all influenced differently by perturbations of the INP scaling factor and CAP is a strong indication that these parameters and their combination change the distribution of $RH_i$ values between different PPE members.

This is indeed what we find: $RH_i$ distributions in PPE members with both large INP scaling factors and large CAP differ strongly from those with low values of both parameters (Fig. 8 a). When accumulating PPE members where both CAP > 0.8 and the INP scaling factor > 1 (red histogram in Fig. 8 a), we find a $RH_i$ distribution that peaks at approximately $RH_i$ = 100% and is slightly skewed to larger values (tapers off at around $RH_i$ = 120%). However, the distribution of $RH_i$ in PPE members





with CAP < 0.4 and an INP scaling factor < 0.5 (cyan histogram in Fig. 8 b) has a much sharper peak at $RH_i$ slightly below 100% and a much larger tail to large $RH_i$ of 135% and more. The mean and median for the first group are both approximately 103%, whereas for the second group the mean of 106 % is larger than the median of 103 %. This explains both the differing behaviours of the mean and median $RH_i$ with CAP and INP as well as the spread, which increases with the asymmetry of the distribution. We note that the peak $RH_i$ for small CAP and INP indicates that most trajectories in those PPE members are subsaturated at the end of the ascent. As we will see in Section 5, ice crystals sublimate in the outflow because of this.




**Figure 7.** Scatter plots of $RH_i^{\mathrm{median}}$ over CAP (a), INP (b) and CCN (c) perturbations, with $25^{\mathrm{th}}$-$75^{\mathrm{th}}$ percentile range shaded in dark grey and $5^{\mathrm{th}}$-$95^{\mathrm{th}}$ percentile range shaded in light grey. Cyan and red points are masked according to INP in a) and CAP in b), with lines indicating a linear fit (can seem curved due to logarithmic axes) through the corresponding colored data points. The same in d), e) and f) except for mean $\tau_{\mathrm{sat,ice}}$.





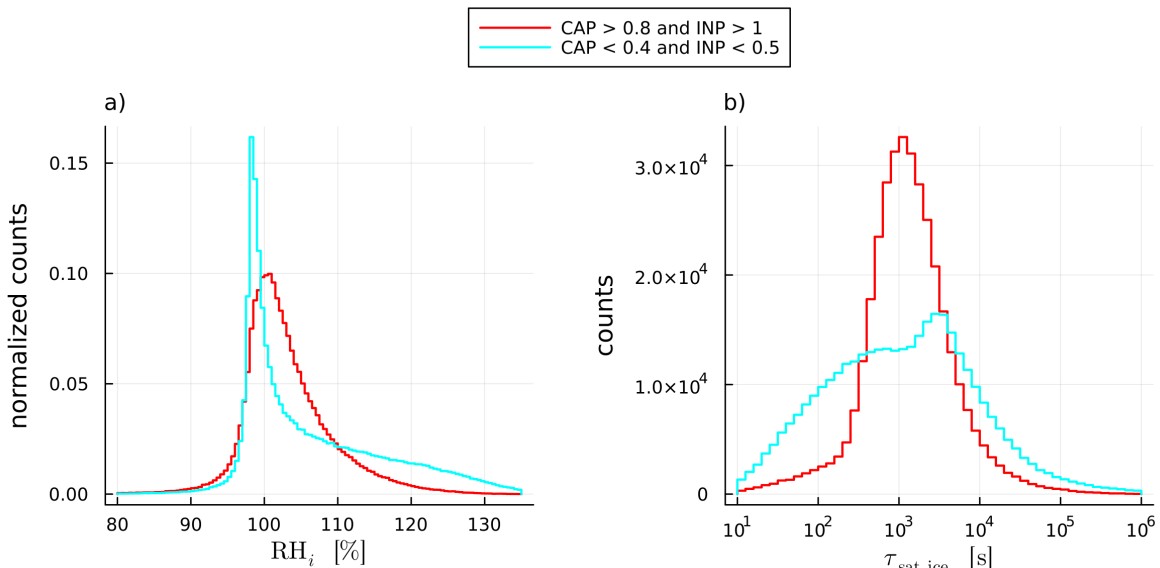

**Figure 8.** Histograms for a) $RH_i$ and b) $\tau_{\mathrm{sat,ice}}$ at the end of the ascent for all PPE members with CAP > 0.8 and INP scaling factor > 1 (red; in total 9 PPE members and 357,888 trajectories) or CAP < 0.4 and INP scaling factor < 0.5 (cyan; in total 9 PPE members and 363,512 trajectories).

Before interpreting these findings, it is essential to note that in the ICON model, supersaturation with respect to **water** can only develop when an air parcel is fully glaciated, as this is the point at which the saturation adjustment scheme, which enforces relative humidity over water of 100%, becomes inactive. Consequently, when examining the effects of parameter perturbations on $RH_i$ at the end of ascent, it is important to focus on the timing of the air parcel's full glaciation, as well as the cloud microphysical evolution after this point, since only when there is no more liquid water will we see deviations from $RH_w = 100\,\%$. Note that $RH_w = 100\,\%$ also constrains $RH_i$ to a temperature dependent but otherwise fixed value (with $RH_i > 100\,\%$). For a fully glaciated air parcel, the relaxation timescale for supersaturation over ice (at a vertical velocity $w = 0\,\mathrm{m\,s^{-1}}$) is inversely proportional to the ice number concentration times the radius of ice (e.g. Khvorostyanov, 1995; Khvorostyanov and Sassen, 1998; Khvorostyanov et al., 2001):

$$\tau_{\mathrm{sat,ice}} = \frac{1}{4\pi \cdot \mathrm{CAP} \cdot N_i \cdot r_i \cdot D_v} \quad (\text{with} \quad [N_i] = \mathrm{m^{-3}}), \tag{2}$$

with $D_v$ the water vapor diffusion coefficient (taken as a constant value of $3 \cdot 10^{-5}\,\mathrm{m^2\,s^{-1}}$). This means that parameter perturbations that lead to higher $N_i$ and $r_i$ should lead to a faster reduction of supersaturation over ice, and therefore, given a similar vertical velocity distribution, on average smaller $RH_i$. This assumption is verified by the two-dimensional histogram of all $RH_i$ and $\tau_{\mathrm{sat,ice}}$ values (taken at the end of the ascent, because $w$ must be approximately 0) from the PPE (Fig. 9). It reveals that when $\tau_{\mathrm{sat,ice}}$ is smaller than approximately $5 \cdot 10^2\,\mathrm{s}$, $RH_i$ values are concentrated at or slightly below approximately 100%. Above this threshold, the mean, median and spread of $RH_i$ per $\tau_{\mathrm{sat,ice}}$-bin increase monotonically and plateau at $RH_i \approx 115\%$





when $\tau_{\mathrm{sat,ice}} > 10^4$ s. This shows that $\tau_{\mathrm{sat,ice}}$ is the primary physical control of $RH_i$. $RH_i$ at the end of the ascent also does not depend on $RH_i$ at the moment of full glaciation, when the air parcels are still ascending (see Supplement SI4) which solidifies that $\tau_{\mathrm{sat,ice}}$ controls $RH_i$ at the end of the ascent. Therefore, to interpret effect of parameter perturbations on $RH_i$, we must first understand their effects on $\tau_{\mathrm{sat,ice}}$.

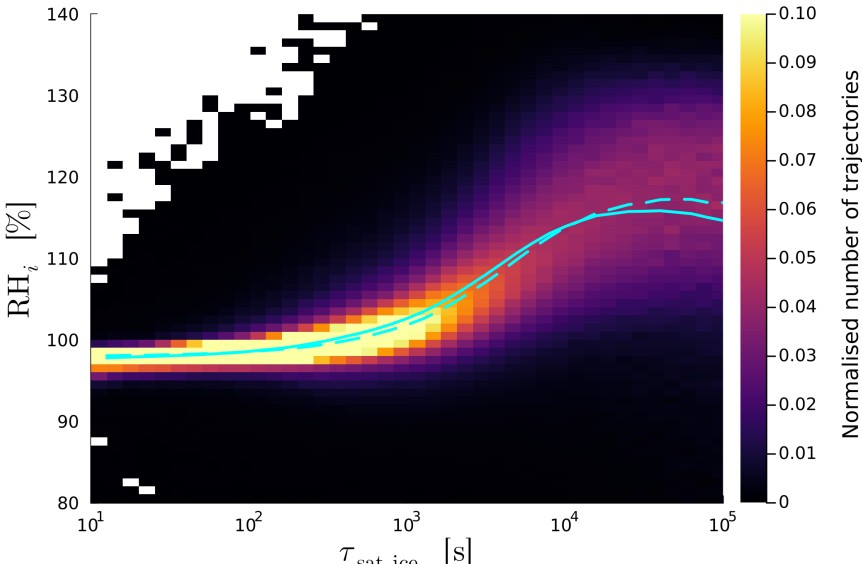

**Figure 9.** Two dimensional normalized histogram of $RH_i$ over $\tau_{\mathrm{sat,ice}}$ for all trajectories from all PPE members. Cyan lines show mean (solid) and median (dashed) values per $\tau_{\mathrm{sat,ice}}$-bin.

The median $\tau_{\mathrm{sat,ice}}$ (Fig. 7 d, e and f, note logarithmic y-axis) varies with CAP, INP and CCN parameter perturbations in a similar manner than $RH_i$ (Fig. 7 a, b and c): CAP reduces $\tau_{\mathrm{sat,ice}}$ when INP scaling is large and increases it when INP scaling is small; the INP scaling factor reduces $\tau_{\mathrm{sat,ice}}$ when CAP is large and increases it when CAP is small; the CCN scaling factor reduces $\tau_{\mathrm{sat,ice}}$ regardless of CAP and INP scaling. As for $RH_i$, joint perturbations in the INP scaling factor and in CAP influence the distribution of $\tau_{\mathrm{sat,ice}}$ (Fig. 8 b). When CAP > 0.8 and INP scaling factor > 1 (red histogram in Fig. 8 b, note the

logarithmic x-axis and logarithmic spacing of bins), $\tau_{\mathrm{sat,ice}}$ values peak at approximately $10^3$ s and fall off symmetrically to either side. On the other hand when CAP < 0.4 and INP < 0.5 (cyan histogram in Fig. 8 b) the distribution is very broad and bi-modal, with the largest peak at a larger value of approximately $5 \cdot 10^3$ s and a second peak at a lower value of approximately $5 \cdot 10^2$ s. As discussed above, $\tau_{\mathrm{sat,ice}}$-values below $5 \cdot 10^2$ s produce almost only $RH_i$-values of approximately 100% (Fig. 9). The abundance of $\tau_{\mathrm{sat,ice}}$-values below this threshold for the second group therefore explains the sharp peak of the $RH_i$ distri-

bution (Fig. 8 a). The larger $\tau_{\mathrm{sat,ice}}$ and higher abundance of trajectories above the threshold also explains the long tail to larger $RH_i$, since $RH_i$ increases strongly with $\tau_{\mathrm{sat,ice}}$ beyond this threshold (Fig. 9).





The effect that the INP scaling factor and CAP have on the distribution of $\tau_{\mathrm{sat,ice}}$ (and thus $RH_i$) can be traced back to their effect on the hydrometeor populations at the end of the ascent (see Sec. 3.2 for physical interpretation), since $\tau_{\mathrm{sat,ice}}$ is

inversely proportional to $N_i$, $r_i$ and CAP (Eq. 2). When the INP scaling factor or CAP is small, the distribution of $N_i$ shows a large spread (Fig. 3,b and e, 4 b and e) and a bulge to larger $N_i$. This in turn creates the bulge to smaller $\tau_{\mathrm{sat,ice}}$ (Fig. 8 b) and ultimately the sharp peak in $RH_i$ (Fig. 8 a). In addition, the larger spread in $N_i$ also means there is a larger abundance of smaller $N_i$ values, which together with the smaller $r_i$ (Fig. 3,c and f, 4 c and f) explains the larger abundance of larger $\tau_{\mathrm{sat,ice}}$-values (Fig. 8 b) and ultimately the long tail to larger $RH_i$ (Fig. 8 a). In contrast, when either the INP scaling factor or CAP

are large, the $N_i$ distribution becomes narrower and more symmetrical, which in turn results in similarly shaped distributions for $\tau_{\mathrm{sat,ice}}$ and ultimately $RH_i$.

The CCN scaling factor decreases the mean and median $\tau_{\mathrm{sat,ice}}$ (Fig. 7 f) but does not change the overall shape and spread of the distribution (see Supplement Fig. SI4). The CCN scaling factor also has no notable effect on the hydrometeor popula-

tions at the end of the ascent (not shown), however the decrease in $\tau_{\mathrm{sat,ice}}$ with increasing CCN scaling (which is weaker than the effect from INP scaling or CAP) must be due to a combined effect of CCN on $N_i$ and $r_i$ that does not appear in our analysis (possibly due to the time resolution of output data).

## 4   Impact of ascent timescale on parameter sensitivity

Schwenk and Miltenberger (2024) found that fast-ascending WCB trajectories transport significantly more hydrometeors into

the UTLS and undergo distinctly different cloud microphysical processes (e.g. more riming, less deposition) compared to slower-ascending ones. Additionally, they demonstrated that precipitation formation pathways in convectively influenced WCB parcels differ markedly from those in slantwise-ascending parcels. Consequently, fast ascending trajectories likely play a key role in modulating moisture transport, which raises the question of whether the effects of parameter perturbations depend on the ascent timescale. In accordance with Oertel et al. (2025), we define trajectories with $\tau_{600} < 10\,\mathrm{h}$ as fast ascending and

repeat the analysis in the proceeding sections for this subset of WCB trajectories. We only briefly summarize how conditions at the end of the ascent change with ascent time, as they are consistent with findings by Schwenk and Miltenberger (2024).

Fast ascending trajectories ascend to lower temperatures and pressures, i.e., to higher altitudes: The average across all PPE members for the mean temperature and pressure are $\sim 230\,\mathrm{K}$ and $\sim 330\,\mathrm{hPa}$, respectively. If only fast ascending trajectories

are considered, these values reduce to $\sim 226\,\mathrm{K} \sim 285\,\mathrm{hPa}$. On average, $q_i^{\mathrm{mean}}$ and $N_i^{\mathrm{mean}}$ are also much larger for fast trajectories ($0.12\,\mathrm{g\,kg^{-1}}$ and $1.6{\cdot}10^7\,\mathrm{kg^{-1}}$) than for all trajectories ($0.04\,\mathrm{g\,kg^{-1}}$ and $0.2{\cdot}10^7\,\mathrm{kg^{-1}}$), and the mean $RH_i^{\mathrm{mean}}$ at the end of ascent for fast trajectories is closer to saturation (100.5%) than for all trajectories (105.7%). The mean $r_i^{\mathrm{mean}}$ is smaller for fast trajectories (0.05 mm) than for all trajectories (0.09 mm). Fast trajectories also glaciate much later (at pressures and temperatures of 360 hPa and 241 K) than all trajectories ( 485 hPa and 253 K,Fig. 10 a and b), which is consistent with the

findings from Schwenk and Miltenberger (2024).





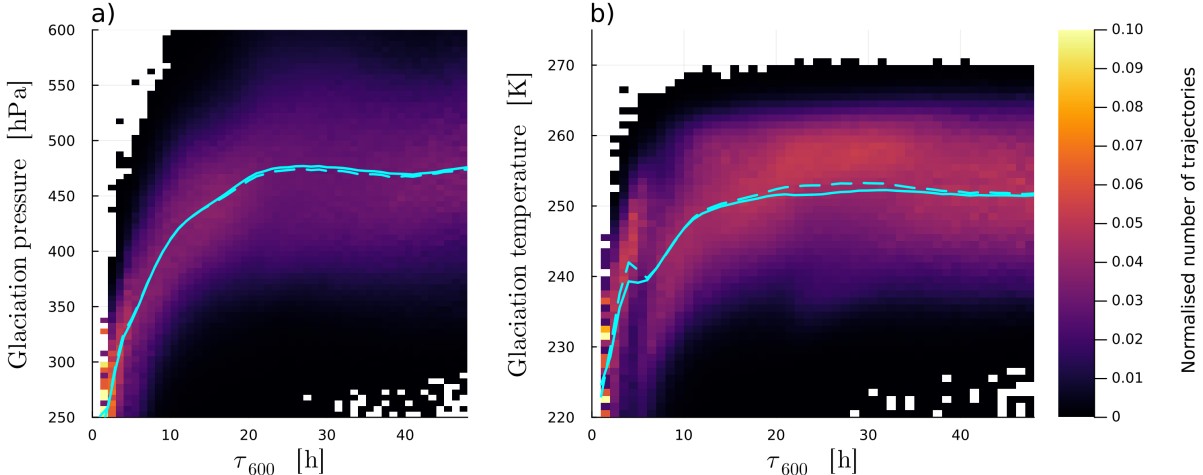

**Figure 10.** 2-dimensional histograms of pressure (a) and temperature (b) at glaciation (defined as the first 60 min time step for which $q_c$ is zero; trajectories with non-zero $q_c$ are excluded) over the fastest 600 hPa-ascent time $\tau_{600}$. Mean and median are shown by the solid and dashed line, respectively.

We first summarize the effects that are similar for fast ascending trajectories and all trajectories. The effects we observed for perturbations in the INP scaling factor (Fig. 4) and CAP (Fig. 3) for all trajectories are also seen for fast ascending trajectories, only that they are much more pronounced (Fig.11 (summary statistics as in Figures 3 and 4 a, b and c are not shown). The largest effect is seen for $N_i$: when the INP scaling factor and/or CAP are small, the second peak towards higher $N_i$ in the bimodal distribution becomes the largest, effectively shifting the peak $N_i$ by two orders of magnitude (from $\sim 10^5\,\mathrm{kg}^{-1}$ to $\sim 10^7\,\mathrm{kg}^{-1}$, Fig. 11 b and e). This means that fast trajectories contain many times more homogeneously freezing cloud droplets, and that this amount increases drastically when the CAP and INP scaling factors are small. $r_i$ also decreases (Fig. 11 c and e) and $q_i$ increases (Fig. 11 a and d) more strongly with decreasing CAP and INP for fast ascending trajectories than for all trajectories. Even though $r_i$ is smaller for fast trajectories (which should increase $\tau_{\mathrm{sat,ice}}$), the much larger $N_i$–values (independent of the choice for INP scaling factor or CAP) lead to substantially smaller $\tau_{\mathrm{sat,ice}}$-values for fast ascending trajectories (compare Fig. 12 b to Fig. 8 b). The large shift in $N_i$ values with INP scaling and CAP also induces a larger difference in $\tau_{\mathrm{sat,ice}}$-values for different INP scaling factors and CAP (Fig. 12 b). However, the effect on the $RH_i$ distribution (Fig. 8 a) is small. This is because the peak $\tau_{\mathrm{sat,ice}}$ for fast trajectories is below $5 \cdot 10^2$ s regardless of INP scaling and CAP, and this value is below the threshold after which $RH_i$ increases above 100 % (Fig. 9. Therefore, fast ascending trajectories reduce ice supersaturation much faster than all trajectories due to their high ice-number concentration $N_i$.

We summarize that for fast ascending trajectories the INP scaling factor and CAP greatly influence the hydrometeor population at the end of the ascent, but that this has only a small effect on $RH_i$ because large very large $N_i$-values lead to small $\tau_{\mathrm{sat,ice}}$ in almost all cases.



**Figure 11.** Histograms of $q_i$ (a and d), $N_i$ (b and e) and $r_i$ (c and f) only for fast ascending trajectories and for subsets of PPE members sorted according to CAP (top row) and the INP scaling factor (bottom row).




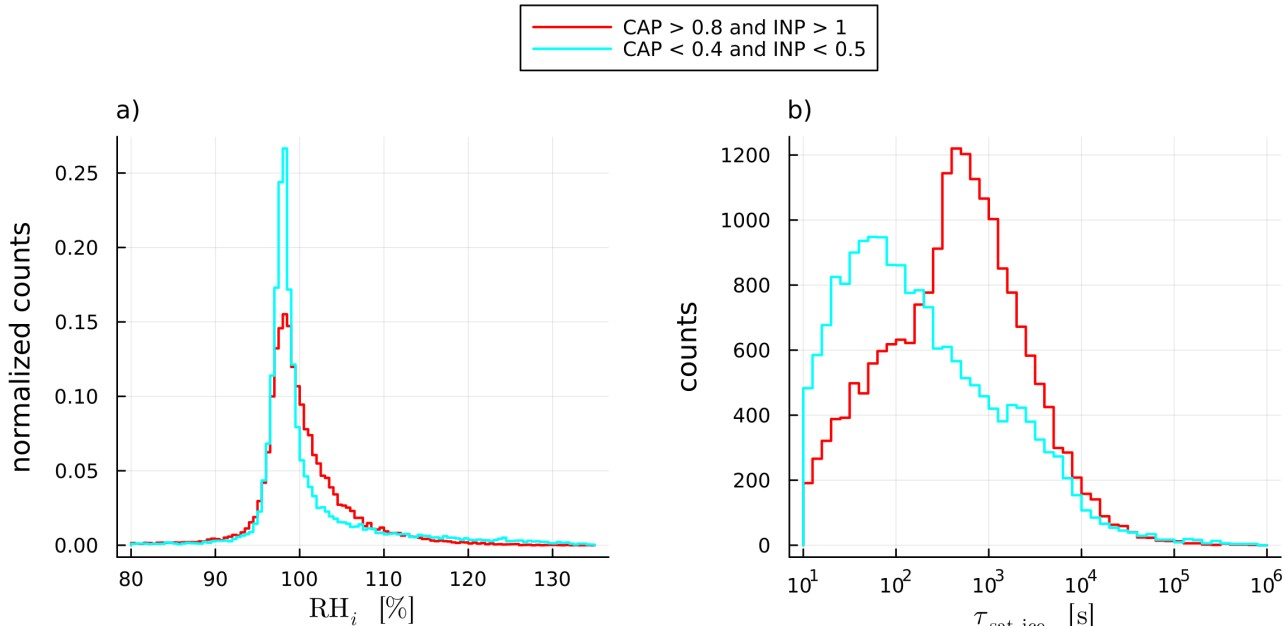

**Figure 12.** Histograms for a) $RH_i$ and b) $\tau_{\mathrm{sat,ice}}$ at the end of the ascent for only the fast ascending trajectories for PPE members with both CAP > 0.8 and INP scaling factor > 1 (red) or CAP < 0.4 and INP scaling factor < 0.5 (cyan).

SAT and the CCN scaling factor have very little impact on the outflow properties of the bulk of WCB trajectories, but become important when considering only the sub-set of fast ascending trajectories. SAT becomes important for the pressure, temperature and specific humidity at the end of the ascent (Fig. 13). While the distributions of these variables do not change visibly with SAT (not shown), the mean and median, as well as the upper percentiles all decrease linearly with an increase in SAT. This indicates that modifications to the saturation adjustment scheme, which permit supersaturation under sufficiently high vertical

velocities, can influence the thermodynamic conditions at the end of ascent when vertical velocities during the ascent are high enough. However, this is also a reflection of the fact that the fast ascending trajectories glaciate much later (Fig. 10), meaning that the saturation adjustment scheme is active for a longer portion of the ascent. Therefore, fast ascending trajectories spend more of their ascent at $RH_w = 100\%$, or slightly above that value if the SAT factor is large. Thus, more vapor, that would otherwise have condensed, is transported higher into the atmosphere, resulting in a delayed latent heat release that provides

some additional buoyancy later in the ascent. However, even for fast trajectories SAT still does not influence $RH_i$. This is because fast ascending trajectories are also mostly glaciated by the end of the ascent, and the $\tau_{\mathrm{sat,ice}}$-values are so small, that supersaturation is quickly reduced regardless of SAT. This means that while SAT can change the pressure and temperature at the end of the ascent for fast ascending trajectories, the specific humidity values remain constrained by the thermodynamic conditions.






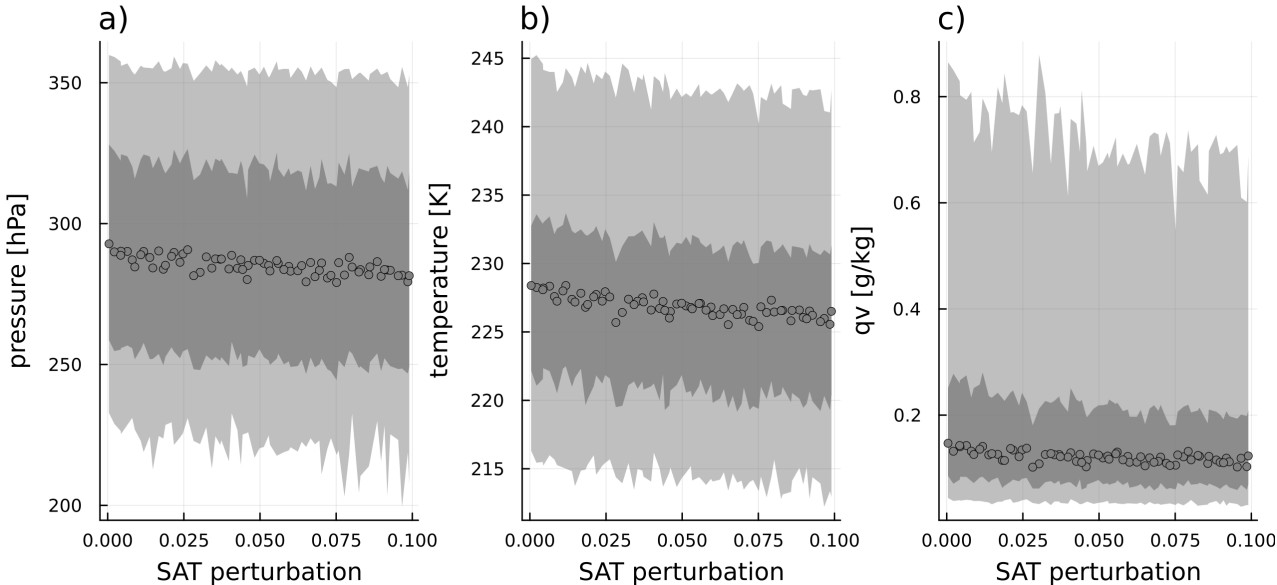

**Figure 13.** Pressure (a), temperature (b) and specific humidity $q_v$ (c) at the end of the ascent for only the fast ascending trajectories over the perturbation of the saturation adjustment scheme (SAT). For plots a) and b) the dots show the mean per PPE member, in plot c) the dots show the median. Shaded areas indicate the inter-quartile range (dark grey) and $5^{\text{th}}$ to $95^{\text{th}}$ percentile (light grey).

For fast ascending trajectories, the CCN scaling factor becomes relevant for $r_i$, $N_i$, $q_i$, $\tau_{\text{sat,ice}}$ and $RH_i$ at the end of the ascent (Fig. 14). However, the shape of distributions is only changed for $N_i$. The effects during the ascent are similar to those discussed in Sec. 3.2 (not shown). With an increase in CCN, the amount of cloud droplets during the ascent increases (Fig. 5 d), and fast ascending trajectories have more cloud droplets that freeze homogeneously (Fig. 11 b and e). Therefore, when the CCN
scaling factor is large and ascent velocities are high, even more cloud droplets freeze homogeneously, decreasing $r_i$ at the end of the ascent (Fig. 14 c) and increasing $N_i$ (Fig. 14 b) as well as $q_i$ (Fig. 14 a). The larger $N_i$ implies smaller $\tau_{\text{sat,ice}}$ (Fig. 14 d) and therefore smaller $RH_i$ (Fig. 14 e), which is on average 102% when CCN scaling < 5 and 99% when CCN scaling > 15. Overall, $RH_i$ changes only slightly across PPE members for fast ascending trajectories, but among these changes, CCN scaling factor perturbations produce the largest effect. This is because for small CCN, $N_i$ is reduced enough to produce $\tau_{\text{sat,ice}}$ values
slightly above the threshold value of $5 \cdot 10^2$ s.

## 5   Effects of parameter perturbations on cirrus properties 5 hours after end of ascent

The conditions at the end of the ascent of a parcel likely influence but are not necessarily representative for those in the hours afterwards. Hydrometeors may precipitate out of the parcel and/or grow to reduce supersaturation (or evaporate in case of sub-saturated conditions). $RH_i$ values 5 h after the end of the ascent have a large spread (not shown) due to the subsidence of a
large portion of trajectories (37 % of trajectories across all PPE members have a pressure at least 10 hPa higher than at the end



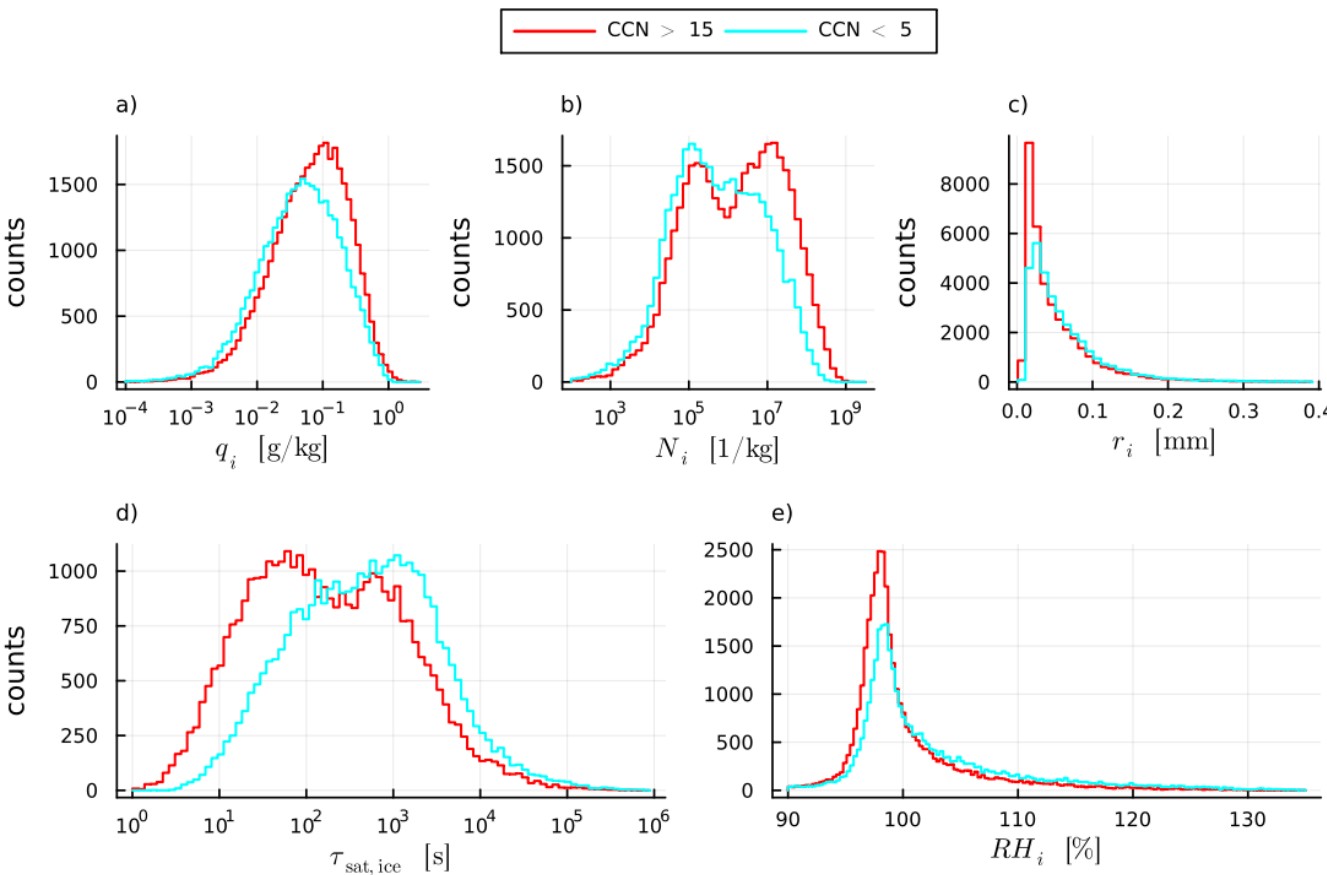

**Figure 14.** Histograms for $q_i$ (a), $N_i$ (b), $r_i$ (c), $\tau_{\mathrm{sat,ice}}$ (d) and $RH_i$ (e) for all trajectories from PPE members with a CCN scaling factor larger than 15 (red; 17 PPE members and 686,947 trajectories) or smaller than 5 (cyan; 17 PPE memebers and 627,912 trajectories)

of the ascent). We are interested in those that could retain large $RH_i$ and therefore consider only those that have no net-descent after 5 h ($\sim 50\,\%$ of all WCB trajectories across all PPE members). For this subset of trajectories, we consider the impact of the parameter perturbations on $q_i^{\mathrm{mean}}$, $N_i^{\mathrm{mean}}$ and $RH_i^{\mathrm{mean}}$ 5 h after the end of ascent. The two most important parameters according to the IBF score are still CAP and the INP scaling factor (Tab. 3) and the $\mathrm{RF}_{\mathrm{X}}^2$-predictions are very good, with $R^2$

and NRMSE of approximately 0.8-0.9 and 0.3-0.5, which supports this statement.

Apart from the deposition nucleation, the INP scaling factor only effects processes in the mixed-phase. Therefore, as long as there is a negligible amount of deposition nucleation, any effect seen from it after the ascent is a reflection of the conditions at the end of the ascent. $N_i$ 5 hours after the ascent increases with the INP scaling factor (Fig. 15 a), which is probably a

reflection of the behavior found at the end of the ascent (Fig. 4 e). However, at the end of the ascent and for small INP scaling factors, $N_i$ has a bimodal distribution, which is no longer seen 5 hours afterwards. One would presume that trajectories with





large $N_i$ and small $r_i$ at the end of the ascent retain the ice for a long time due to the lower sedimentation velocity of smaller ice crystals. However, the fact that it does not lead to higher median $N_i$ 5 h after the ascent means that these small ice crystals are most likely evaporated. Indeed, only the subset of trajectories with $RH_i$ close to 100% 5 hours after the ascent still have the bimodal distribution in $N_i$ that supports $N_i$ values larger than $10^6 \, \mathrm{kg}^{-1}$ (see Supplement Fig. SI6). Alternatively, some small ice crystals also persist when CAP is high (Fig. 15 b), which would explains why CAP is more important for $N_i^{\mathrm{mean}}$ 5 h after the ascent even though the peak $N_i$ is not shifted.

For $RH_i$ CAP and the INP scaling factor are almost equally important (Tab. 3). They also have a very similar effect of increasing the tail of $RH_i$-values 5 h after the ascent to larger values while leaving the peak at $\sim 100\%$ unchanged (Fig. 15). For the INP scaling factor this is a reflection of different $N_i$ 5 h after the ascent (smaller $N_i$ means slower reduction of supersaturation) and for CAP the signal simply occurs because larger CAP values reduce supersaturation faster.

|        | mean($RH_i$) | mean($q_i$) | mean($N_i$) |
|--------|--------------|-------------|-------------|
| CAP    | 0.291        | 0.540       | 0.524       |
| INP    | 0.461        | 0.206       | 0.266       |
| CCN    |              |             |             |
| SST    | 0.109        | 0.103       |             |
| SAT    |              |             |             |
| $R^2$  | 0.849        | 0.833       | 0.758       |
| NRMSE  | 0.385        | 0.406       | 0.488       |

**Table 3.** Impurity-based feature (IBF) importance scores of CAP, INP and CCN for $RH_i^{\mathrm{mean}}$, $q_i^{\mathrm{mean}}$ and $N_i^{\mathrm{mean}}$ 5 h after the end of ascent. Only scores over 0.1 are included. The bottom rows show the $R^2$ and NRMSE value for the predictions of the forest regression model $\mathrm{RF}_X^2$. Note: these values are slightly different each time a forest model regression is performed; the values shown here are averaged over 100 iterations.

## 6 Conclusions and Discussion

Warm-conveyor belts are a key feature for extra-tropical transport of humidity from the lower troposphere to the tropopause region. Due to the strong ascent and associated cloud formation this transport is modulated by cloud physics processes, the representation of which suffers from well-documented uncertainties in state-of-the-art weather and climate models. In this paper we therefore analyze Lagrangian trajectory data from a perturbed parameter ensemble (PPE) to answer the following research questions: (How) do perturbations in microphysical parameters and SST (i) influence thermodynamic and moisture conditions in fresh WCB outflow, (ii) alter hydrometeor distributions in fresh WCB outflow, (iii) impact the moisture and ice properties in aged WCB outflow, and (iv) how these effects vary with ascent velocity?



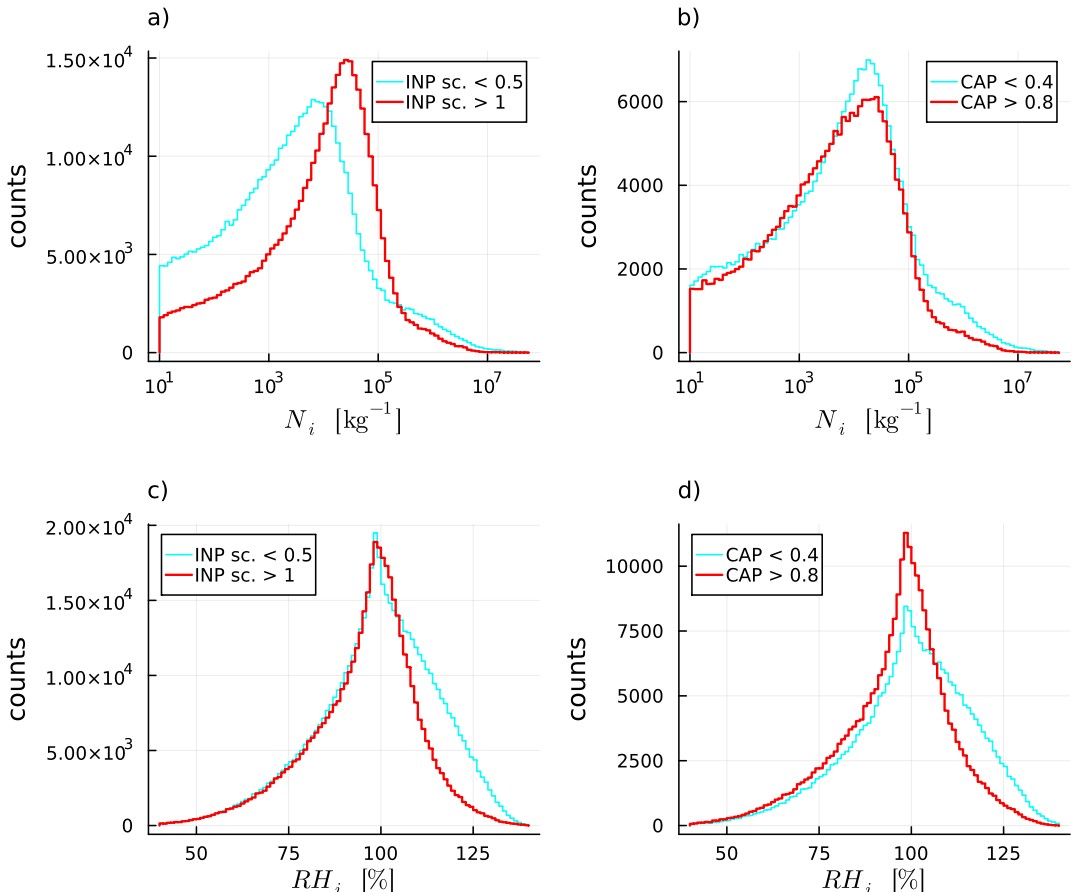

**Figure 15.** Histograms for $N_i$ 5 h after the end of ascent, split according to the INP scaling factor in (a) and according to CAP in (b). The same for $RH_i$ in (c) and (d). Only trajectories that do not descend after the end of the ascent are included.

The distribution of thermodynamic properties and specific humidity at the end of the ascent, i.e. $T$, $p$ and $q_v$, (**RQ i**) hardly varies across PPE members (Fig. 2 a-f). This indicates that the thermodynamic conditions at the end of the ascent are constrained by the larger-scale environment and that parameter perturbations do not substantially influence their magnitude or distribution. In contrast, the $RH_i$ distribution and its mean value varies quite strongly with INP scaling factor and CAP: When the INP scaling factor and CAP are small, the $RH_i$ distribution has a sharp peak below 100% and a long tail to larger values (Fig. 8 a). If the INP scaling factor and CAP are large, the distribution is more Gaussian-shaped with a peak at about 106 %. We have shown that the differences in the $RH_i$ distribution are caused by the impact of INP scaling factor and CAP on the ice crystal population, which is connected to $RH_i$ by the timescale for super-saturation depletion $tau_{\mathrm{sat,ice}}$(Eq. 2, Fig. 9).

The hydrometeor population at the end of ascent (**RQ ii**) is dominated by ice particles. The largest variation in ice mass and number concentration are induced by the INP scaling factor and CAP: For a small CAP and INP scaling factor, the $N_i$





distribution is bimodal, while the second peak at large $N_i$ values vanishes for larger CAP and INP scaling factors. This behavior can be explained by the more efficient cloud glaciation with large CAP values and INP scaling factor, which strongly reduces the number supercooled droplets reaching the homogeneous freezing level. For enhanced INP scaling factors we additionally detect a small shift in the lower $N_i$, which correlates directly with the number of heterogeneously formed ice crystals. The ice crystal mass mixing ratio decreases with increasing CAP (and to a lesser extend with increasing INP scaling factor) due to the

larger size of ice crystals and more efficient sedimentation.

Interestingly we find no significant impact of CCN scaling on the ice crystal population at the end of ascent, although it substantially modulates the hydrometeor population and onset of substantial glaciation during the ascent. However, this signature does not propagate to the fully glaciated state at the end of ascent. Also we do not find an impact of SST or SAT

perturbations on the outflow ice crystal population. This is somewhat surprising, as Oertel et al. (2025) find altered fractions of fast ascending trajectories and Schwenk and Miltenberger (2024) showed that fast ascending trajectories contain more ice at end of ascent. However, the fraction of fast ascending trajectories is overall very small and therefore the impact on the overall distribution of $q_i$ and $N_i$ in the outflow are small.

Considering only fast-ascending trajectories (**RQ iv**) CAP and the INP scaling factor remain the most important parameters for hydrometeor content and $RH_i$ at the end of the ascent (Figs. 11 b and e, Fig. 12 b), albeit with the a less pronounced impact on the $N_i$ and $RH_i$ distributions than for all trajectories. This is likely a consequence of the fast ascent and short timescale for glaciation with some super-cooled liquid reaching the homogeneous freezing temperature irrespective of the choice of CAP and INP scaling factor. In contrast to the consideration of all trajectories, perturbations of the CCN scaling factor have a small

impact $q_i$, $N_i$ and $RH_i$ distributions with large CCN scaling factors shifting those distributions to smaller values. We also find a small impact of SAT perturbations on the thermodynamic conditions ($p$, $T$) and specific humidity $q_v$ at the end of ascent (Fig. 13). The larger impact of SAT for fast-ascending trajectories is due to more significant super-saturation being possible at larger vertical velocities and the larger fraction of the ascent spend in mixed-phase conditions, where water vapour-condensate partitioning is controlled by saturation adjustment. Therefore, altering SAT impacts buoyancy more strongly and enables a

deeper ascent with an increasing SAT perturbation.

The ice particle population 5 h after the end of the ascent (**RQ iii**) is impacted mostly by the choice of CAP and the INP scaling factor. In comparison to end of ascent (RQ ii), the large $N_i$ values (associated with small $r_i$) are not present anymore and crystals from the small $N_i$ mode dominate. Consequently the INP scaling factor is more important than CAP 5 h after the

end of ascent, while they were of similar importance at the end of ascent.

The analysis presented here is based on the PPE from Oertel et al. (2025), which focused on the dynamical impact of parameter perturbations, i.e. on the role for diabatically induced potential vorticity anomalies. Some earlier studies have also provided insight into the role of uncertainty in the microphysical parametrization for WCBs, but in contrast to our paper



have predominantly focused on the impact on diabatically induced PV anomalies, upper-level flow and its predictability. In particular, Joos and Forbes (2016) investigated differences between two versions of the ECMWF global model (IFS) dominated by changes in the cloud microphysics scheme. Similarly, Mazoyer et al. (2021) and Mazoyer et al. (2023) explored the impact of using different cloud microphysics schemes available in the French Mesoscale Non-Hydrostatic model (MesoNH) in convection-permitting simulations. Pickl et al. (2022) investigated the effect of Stochastically Perturbed Parameterization

Tendencies (SPPT) on WCBs using the IFS model at 100 km resolution, running simulations over multiple model years. In addition, recent studies have proposed a concept for objectively selecting impactful parameter perturbations Hieronymus et al. (2022); Neuhauser et al. (2023). While their results partly align with the subjective parameter selection in our PPE, guidance from their results could be used to expand the investigation in future studies.

Our results suggest that in ICON's two-moment cloud microphysics scheme, the hydrometeor populations and $RH_i$-conditions in WCB outflow are sensitive to the parameter choices in the cloud microphysics scheme, specifically those relevant for microphysical processes in the ice phase (CAP and INP scaling factor). The findings from this study could be important for multiple reasons:

– The effects identified in this study could inform decisions for microphysical parameter choices in numerical models or
parameters to be perturbed in ensemble prediction systems.

– Comparison of the distribution of $N_i$, $q_i$, $r_i$ and $RH_i$ documented here and distributions from large observational datasets, may help in constraining realistic values for CAP and INP scaling factor for WCB flow configurations.

– Future modeling studies on WCB moisture transport should keeo CAP and the INP scaling factor, but reserve additional resources to investigate other potentially important factors such as the representation of graupel and ice sedimentation
velocity or collision efficiencies of frozen particles.

– Our findings may have implications for geoengineering strategies such as cirrus cloud thinning (Lohmann and Gasparini, 2017), which aim to modify processes similar to those affected by the INP scaling factor. Since $N_i$ 5 h increases with the INP scaling factor 5 after the ascent, accelerating these processes could pose a risk of 'overseeding', as discussed in Gasparini et al. (2020).

– The vertical water transport in WCBs is an important contribution to the water content and supersaturation distribution in the extra-tropical UTLS. Future studies should examine the impact of the changes found here for the occurrence of extra-tropical ice supersaturated regions, which impact downstream in-situ cirrus formation including aviation-induced cirrus.

*Code and data availability.* Code will be made available after acceptance but can be showed to reviewers if desired. WCB trajectory data
for the 70-member PPE are publicly available in RADAR4KIT https://www.doi.org/10.35097/ecgs4f56mp3ymjmt.





## Appendix A: Additional Figures

**Figure A1.** a) scatter-plot of $95^{\text{th}}$ percentiles of $q_v$ over $T$ per ensemble member (dots). The ensemble members with the highest (lowest) $95^{\text{th}}$ percentile of $q_v$ values are plotted in cyan (orange). b) scatter-plot of saturation specific humidity (calculated using temperature and pressure of parcel) against specific humidity $q_v$ for these two ensemble members. c) shows the Spearman correlation of mean and $95^{\text{th}}$ percentile for $q_v$ and $T$ with all perturbed parameters.





*Author contributions.* AO set up and conducted the simulations for the PPE and provided the pre-selected Lagrangian WCB trajectory data. AM and AO developed the concept of this study. CS conducted/performed the post-processing calculations, the statistical analysis, and visualized the results. All three authors were involved in the discussion and interpretation of the results.

*Competing interests.* One author is a member of the editorial board of ACP (Annika Oertel).

*Acknowledgements.* The PPE was built within subproject B8 of the Transregional Collaborative Research Center SFB / TRR 165 "Waves to Weather" (www.wavestoweather.de) funded by the German Research Foundation (DFG). AO is supported by the Italia–Deutschland science-4-services network in weather and climate (IDEA-S4S; INVACODA, grant no. 4823IDEAP6). This Italian–German research network of universities, research institutes, and the Deutscher Wetterdienst is funded by the BMDV (Federal Ministry of Transport and Digital 715  Infrastructure). The authors acknowledge support by the state of Baden-Württemberg through bwHPC. The PPE simulations were carried out on the supercomputer HoreKa at Karlsruhe Institute of Technology, Karlsruhe, which is funded by the Ministry of Science, Research and the Arts Baden-Württemberg, Germany, and the German Federal Ministry of Education and Research. This work was also funded by the Deutsche Forschungsgemeinschaft (DFG, German Research Foundation) – TRR 301 – Project-ID 428312742: "The tropopause region in a changing atmosphere", sub-project B08 coordinated by Annette Miltenberger.



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
