# Peer review of "Microphysical Parameter Choices Modulate Ice Content and Relative Humidity in the Outflow of a Warm Conveyor Belt"

_EGUsphere, 2025_

## Referee Comment (RC1)

Using a perturbed parameter ensemble (PPE), this interesting study investigates how microphysical uncertainties affect water transport and cirrus properties in a warm conveyor belt (WCB) case. While thermodynamic conditions at the end of ascent are largely insensitive to parameter changes, ice content and relative humidity show strong variability, primarily controlled by ice crystal capacitance (CAP) and ice-nucleating particle (INP) scaling. These sensitivities are especially pronounced in fast-ascending parcels, where modifications to the saturation adjustment scheme further affect cloud properties.

Overall, the manuscript is logically structured, clearly written, and suited for publication in ACP. I recommend acceptance after minor revision. My specific comments and questions are outlined below, followed by a few technical remarks and suggestions for improvement.

**Major Comments**

- In Oertel et al. (2025), and in this paper, a multidimensional Gaussian Process (GP) emulator is used to approximate model behavior. This approach assumes that input parameters, such as capacitance, vary smoothly and that the resulting model output — including variables like ice number concentration and mean ice particle size — responds in a continuous and predictable way.

  Although internal nonlinearities exist in the Seifert and Beheng (SB, 2006) microphysics scheme (e.g., mass–size relationships, category transitions), these are embedded within the model physics and do not necessarily introduce discontinuities in the macroscopic output variables if each hydrometeor category is analyzed separately.

  However, imposed minimum or maximum limits on ice number concentration may lead to plateaus or discontinuities in mean ice size, particularly when ice mass continues to grow while the number concentration is constrained. Additionally, ICON's saturation adjustment scheme limits supersaturation to 100% in regions containing cloud droplets or rain, potentially producing flat regions in the relative humidity w.r.t. water output fields. Given that a Gaussian Process emulator is applied, do these plateaus or discontinuities introduce challenges (if they exist in any of the output fields) for the emulator's ability to accurately capture the underlying parameter–response relationships?

- In these ascending parcels considered within the WCB, the updraft speed is not zero. For example, when $w = 0$ and $N_i r_i = 10^{-1}\,\mu\mathrm{m\,cm^{-3}}$, the relaxation time can be as large as $10^4\,\mathrm{s}$ (Fig. 1). If $w = 1\,\mathrm{m/s}$, the relaxation time can be reduced by approximately an order of magnitude under upper tropospheric conditions. For larger values of $N_i r_i > 10^1\,\mu\mathrm{m\,cm^{-3}}$, the impact of $w \leq 1\,\mathrm{m/s}$ becomes negligible (e.g., Korolev and Mazin (2003)).

  Assuming $w = 0$ for all analyses of ascending parcels may lead to severely overestimated the ice phase relaxation times, particularly for fast-ascending trajectories.

It is stated in Line 492 that $w \approx 0$ for parcels taken at the end of ascent, but later, for example in Section 4, fast ascending trajectories are analyzed together with relaxation timescales. It is therefore recommended to explicitly include $w$ in the relaxation timescale formulation to make this analysis more robust.

– Line 430: "The mean maximum mixing ratio for snow (max(qs)) during ascent also increases with CCN (Fig. 5 e), presumably because in the two-moment microphysics scheme riming contributes to the mass growth rate of snow,...". The reasoning here may not agree with how riming affects particles in SB. In this case: When riming occurs it can have a two fold effect on ice and snow. If ice or snow rimes with raindrops it is converted to graupel else if it rimes with cloud droplets it remains either ice or snow unless it exceeds a critical rime mass threshold depending on some tuning parameter. Typically snow is considered as pristine and therefore is converted quite rapidly to graupel (by assuming a low space filling constant). If riming on snow occured more often then qs would most likely decrease because of this conversion, not increase, right?

I would suggest the process look more like the following: Increased CCN concentrations lead to more numerous but smaller cloud droplets, which slow the formation of raindrops. This, in turn, reduces the collision efficiency between ice or snow particles and cloud droplets, slowing the conversion to graupel. As a result, snow remains more abundant.

– Line 394: I follow this reasoning, but a similar reasoning can be applied to lower INP scaling: with fewer activated INPs, there is less competition for available vapor, resulting in larger mean ri. This in turn enhances the conversion from qi to qs through enhanced collisions with raindrops and ice particles. Some aspects of this or the authors reasoning may become clearer, and confirming the authors explanation, when qs is also plotted in reference to Fig. 4d, e, and f.

**Minor Comments**

– Line 27: Instrument uncertainties remain, with calibrated measurements typically accurate to within 5–10% in supersaturated regimes (Petzold et al., 2020). Can it be added to highlight uncertainty with instruments.

– Line 42: "tropopause region has increased on average from 2011 to 2020 compared to the 1980s". Can a more specific value be added?

– Line 48: "key contributor to extratropical UTLS moisture". By how much do WCBs contribute?

– Line 59: "various atmospheric constituents". Be more specific and replace the phrase with ...water vapor, hydrometeors and aerosols...

– Line 64: What is meant by the "incorrect representation of WBCs"? What is misrepresented? The location, vertical extent, cloud dynamics, cloud microphysics or water vapor transport?

– Line 70: Cloud overlap can also change a cooling effect into a warming effect during the day (Johansson et al., 2019).

- Line 73: Is the warming effect the average when considering day (cooling) and night (warming)?

- Line 105: "Supersaturation" is a continuous variable. Change the sentence to: "otherwise, greater or more widespread supersaturation would be produced."

- Line 118: If the cloud is glaciated then CCN becomes less relevant because of the Wegener-Findheisen-Process. In this case would CCN matter? Before glaciation CCN may be very important.

- Line 165: I realize that is given in previous papers, but it would be good to have a very short one paragraph overview here with Figure 1 from Oertel et al. (2025).

- Line 219: Replace the sentence with "In the PPE, the scheme is modified to allow supersaturation to develop under conditions of strong vertical velocity."

- Line 394: I follow this reasoning, but a similar argument applies to lower INP scaling: with fewer activated INPs, there is less competition for available vapor, resulting in larger mean $r_i$. This in turn enhances the conversion from $q_i$ to $q_s$. Some aspects of this reasoning might become clearer, and confirming the authors explanation, when $q_s$ is also plotted in reference to Fig. 4d, e, and f.

- Figure 3 caption: Keep the caption between Fig 3 and 4 consistent... is it median $q_i$, median $N_i$ median $r_i$ or only $q_i$, $N_i$, $r_i$? I believe it should Fig. 4's description is incorrect.

- Line 444: I don't fully understand this hypothesis. Could you please provide one or two additional clarifying sentences? Does it suggest that stronger convection over higher SSTs would lead to increased $N_i$ and $q_i$? If so, it seems that the primary driver would not be the SST itself, but rather the associated increase in vertical motion.

- Line 548: Issue with brackets.

- Line 551: It is a interesting idea. Can you plot the cloud droplets numbers concentration in each case just before the homogeneous freezing temperature? E.g. are there more cloud droplets available for CAP < 0.4 and INP scaling < 0.5? If so, this will solidify your reasoning.

- Line 559: Issue with brackets.

- Line 593: Ice hydrometeors sublimate. "...(or evaporate/sublimate in case of...".

- Line 508: If one further assumes that the parcel is subsaturated, or that collisional growth is negligible and no conversion to snow occurs under supersaturated conditions, right?

- Table 3. Keep the caption consistent with Table 2 where you can.

- Figure 15 and others. Consistency. Sometimes a) is used or (a)

[Figure]

**Figure 1.** Phase relaxation timescale dependence on updraft speed and Ni*Ri. Solid lines are for an updraft speed (Uz) = 0 m/s and the dashed lines are for the corresponding Uz of 1 m/s. The figure follows the equations from Korolev and Mazin (2003) and extent it to different atmospheric conditions.

- Line 629: Define (RQ) above where Research Question is mentioned.

- Line 634: Consistency with using tau or $\tau$.

- Line 692: How would one constrain realistic values for CAP if it is a function of the habit of ice crystals which keeps changing depending on the state of the environment?

- Line 693: "Keeo". Keep

- Line 698: "scaling factor 5 h after".

- Line 704: In the intoduction Earth's Radiation budget is mentioned and the sensitivity to vapor content in the UTLS. How does the results presented here have an impact radiation.

**References**

Johansson, E., Devasthale, A., Ekman, A. M. L., Tjernström, M., and L'Ecuyer, T.: How Does Cloud Overlap Affect the Radiative Heating in the Tropical Upper Troposphere/Lower Stratosphere?, Geophysical Research Letters, 46, 5623–5631, https://doi.org/10.1029/2019GL082602, _eprint: https://agupubs.onlinelibrary.wiley.com/doi/pdf/10.1029/2019GL082602, 2019.

Korolev, A. V. and Mazin, I. P.: Supersaturation of Water Vapor in Clouds, Journal of the Atmospheric Sciences, 60, 2957–2974, https://doi.org/10.1175/1520-0469(2003)060<2957:SOWVIC>2.0.CO;2, 2003.

Petzold, A., Neis, P., Rütimann, M., Rohs, S., Berkes, F., Smit, H. G. J., Krämer, M., Spelten, N., Spichtinger, P., Nédélec, P., and Wahner, A.: Ice-supersaturated air masses in the northern mid-latitudes from regular in situ observations by passenger aircraft: vertical distribution, seasonality and tropospheric fingerprint, Atmospheric Chemistry and Physics, 20, 8157–8179, https://doi.org/10.5194/acp-20-8157-2020, publisher: Copernicus GmbH, 2020.

Seifert, A. and Beheng, K. D.: two-moment cloud microphysics parameterization for mixed-phase clouds. Part 2: Maritime vs. continental deep convective storms, Meteorology and Atmospheric Physics, 92, 67–82, https://doi.org/10.1007/s00703-005-0113-3, 2006.

---

## Author Comment (AC1)

**Response to RC3**

We thank the reviewer for their comments and very positive feedback on the quality of our work.

Below, we list our responses to their comments and suggestions. More changes than listed have been implemented to accommodate the suggestions by the other three reviewers.

**SPECIFIC COMMENTS**

The "ice" and "snow" categories are very important in this study. First, these should be clearly defined in the paper; these are categories of small and large unrimed ice crystals in bulk and bin microphysics (mp) schemes. In traditional bulk schemes such as ICON's SB two-moment scheme, these ice-phase categories are predefined and with prescribed physical properties (e.g. capacitance, mass-fall speed parameters) and have the necessary but purely artificial process of conversion between categories. In nature, there is no such thing as "conversion from ice to snow". In traditional category-based scheme, the way this process is "parameterized" ultimately impacts the relative distribution of ice and snow, subsequently impacting the mp growth rates and distribution of hydrometeor mass.

With this in mind, how general are your conclusions with respect to the mp scheme used? Would you expect the same results/conclusions using a different category-based mp scheme (with different ice-to-snow conversion) or with a property-based scheme (like P3) which uses generic ice-phase categories with no artificial conversion? I think it would be useful to add some discussion (e.g. in the Conclusion section) on this topic.

We focus our analysis on hydrometeors and vapor in the UTLS at the end of the WCB ascent, where virtually all hydrometeors are ice. The amount of snow at the end of the ascent is very small ($q_s$ is on average $10^{-3}$ g/kg and $N_s$ on average 500 1/kg, compared to $5*10^{-2}$ g/kg for $q_i$ and $2.5*10^6$ 1/kg for $N_i$).

Therefore, at and after the end of the ascent, we do not look at snow. When we talk about snow during the ascent, we use the definitions from the Seifert and Beheng (2006) microphysics scheme, which is referenced in the method section. It is not common to define the hydrometeor species exactly when making use of well-known and widely used physical schemes.

The point on the choice of microphysics scheme is important, but apart from pure speculation, we are not able to determine what differences one would see in such a complex system as a WCB when another scheme is used. However, to address this point, we add the following sentence in the conclusion (Line 750):

*"However, this study consists of only one case study using one microphysics scheme, and it isn't clear how representative our findings are of WCBs (or other model configurations and microphysics schemes) in general."*

**MINOR POINTS**

1. The description of saturation adjustment in ICON (line 216) seems not quite right. Presumably supersaturation with respect to ice can remain. This seems to be better explained on line 479, but perhaps it should be clarified earlier.

This is a good point. We have made clear that the saturation adjustment scheme is designed to produce 100% relative humidity over **water**. The beginning of the paragraph now reads (Line 240):

*"As discussed in the introduction, when a cloud contains liquid water, the saturation adjustment scheme used in ICON instantaneously removes supersaturation or in-cloud sub-saturation by condensing or evaporating excess or deficient water vapour until a relative humidity of 100% over water is achieved."*

2. Line 315: If the units for ice number are in # kg^-1, this is "number mixing ratio", not number concentration. Concentration units in # m^-3 (as correctly indicated in eqn. (2)).

Thank you for pointing this out. We have changed "number concentration" to "number mixing ratio" where it is appropriate.

3. 5: In the first color bar, the two dark red colors for "max_qc" and "max_qs" are very hard to distinguish.

The colors in the bottom figure are not labels, but indicate the spearman correlation coefficient for the correlation of max_qX with the CCN scaling factor, given the colorbar below. That the colors match very well from left to right with the colorbar is a coincidence. If max_qc and max_qs have similar spearman correlation coefficients, then they have a similar color.

4. The paragraph starting on line 671 seems more like it should belong in the Introduction section.

This paragraph provides a context for our study within the current body of research, highlighting the novel aspects that distinguish it from similar publications. We have chosen to include it in the section where we discuss our results, which keeps the

introduction focused on the literature overview. Since no other reviewer seems to object, we would prefer to keep this paragraph where it is, but if the reviewer insists, we can move it to the introduction.

---

## Author Comment (AC2)

**Response to RC4**

We thank the reviewer for their comments and extremely positive feedback on the quality of our work, which we greatly appreciate.

In our revised manuscript, we have:
(i) added more context to the introduction to address the additional studies suggested by the reviewer;
(ii) pointed out the novelty of the PPE and our inability to compare it to other results;
(iii) changed the INP fits to work with logarithmically spaced values and clearly stated the significance of each fit using more precise language and listing Spearman correlation coefficients; and
(iv) updated our figures to implement the reviewer's suggestions.

More changes than listed have been implemented to accommodate the suggestions by the other three reviewers. Also, figure numbers are shifted by one due to the incorporation of a new figure depicting a synoptic overview (now Fig. 1). Below we list our answers to the specific comments.

**Responses to specific comments**

Lines 42-48 ("While most of the aforementioned...contributor to extratropical UTLS moisture."): I would like to see more literature on UTLS moisture transport in the extratropics cited, with explanations of how the authors' study relates to this literature. Examples of articles the authors may want to cite include Weigel et al. 2016 ("UTLS water vapour from SCIAMACHY limb measurements V3.01 (2002-2012)"), Heller et al. 2017 ("Mountain waves modulate the water vapor distribution in the UTLS"), and Sun et al. 2017 ("Characteristics of water vapor in the UTLS over the Tibetan Plateau based on AURA/MLS observations").

Thank you for bringing these studies to our attention. We have revised the introduction to provide a more accurate summary of the extent to which studies have investigated moisture transport in the extratropical UTLS. (Lines 46-59)

Table 1: It would help to show in the table whether perturbations for each parameter between the min and max are spaced linearly (e.g., -2, -1, 0, 1, 2 for SST), logarithmically (e.g., 0.01, 0.02, 0.04, 0.08, ..., 5, 10, 20 for INP), or otherwise, as well as the number of distinct values tested for each parameter.

We have added this information to Table 1.

Lines 339-345 ("We interpret this inability...with parameter perturbations."): The authors should try to compare their results to related findings in at least one study besides Oertel et al. (2025) and see whether any explanation for their results can be found in past literature, and if so, whether the explanation given in past literature is plausible or whether a new explanation is needed. If the authors' results contradict past literature, or if no past literature exists for comparision, that too would be important to mention.

The study done by Oertel et al. (2025) is, to our knowledge, the only perturbed parameter ensemble study ever conducted on warm conveyor belts and there is little other systematic information on the impact of microphysical choices for warm conveyor belt moisture transport. Therefore, our findings cannot be compared to previous studies. However, we now point out the novelty and uniqueness of the study more clearly in the introduction.

Lines 384-393 ("We interpret these findings...ice mass mixing ratio qi."): See comment for lines 339-345.

See comment above.

Lines 406-417 ("We interpret...(heavier ice particles fall out more quickly)."): See comment for lines 339-345.

See comment above.

Lines 422-432 ("This clear dependancy...dominate the ice-phase cloud microphysics."): The authors should check to see whether any related findings exist in any past literature. If so, the authors should cite the literature and relate their analysis to the analysis done in the literature. If not, the authors should clarify that their findings and analysis are original.

See comment above.

Lines 460-462 ("The same is true the other way around...(red dots in Fig. 7 b)."): Please see my comment about figures 7b and 7e. I am not convinced that any linear fit between INP and RH (tao_sat) is statistically significant. Once a relationship between INP and RH (tao_sat) is decided upon, the statement in lines 460-462 should be updated.

Figures 7b and 7e: I am not convinced that the linear fits plotted are worthy of publication. As the authors mention, the INP scaling axis is logarithmic, yet the relative humidity and timescale for supersaturation over ice are fit to linear functions of INP

scaling. This creates a situation where only a small fraction of the data has statistical significance in estimating the slope of the fits. Further fuelling my skepticism, the random variability in the median relative humidity and timescale appears to be at least as significant as the total variance captured by the linear fits. If the authors choose to fit relative humidity and timescale for supersaturation over ice to INP scaling, I urge them to consider non-linear fits and test whether any best fit, particularly for any non-linear relationships supported by previous literature, has a significant Spearman correlation coefficient.

This is a very good suggestion. For INP, we have now changed the fit instead to work on the logarithm of the INP scaling factor. In the updated Figure, the line is now straight and more clear.
We revise the text to additionally note the spearman correlation coefficients for these fits, and relativize the significance of the trends (i.e., point out weak correlations), such that the reader is clearly informed on the significance of the correlation and the trends we describe.

Technical comments:

Figure 1: The text in the legend above the subplots should be made larger.

Figure 1c: The text on each axis and on the colorbar should be made larger.

Figure 8: The text in the legend above the subplots and the numbers on each subplot's axes should be made larger.

Figures 9-10: The numbers and text in colorbars and the numbers along axes should be made larger

We have implemented all of these changes concerning the Figures.

---

## Author Comment (AC3)

**Response to RC2**

We would like to thank the reviewer for their valuable comments, their thorough reading of the paper (and supplement), and their recommendation to accept it for publication. The reviewer raised important points and made smart observations that have substantially improved the paper. When we do not implement one of their recommendations, we provide a clear explanation of our reasoning.

More changes than listed have been implemented to accommodate the suggestions by the other three reviewers. Also, figure numbers are shifted by one due to the incorporation of a new figure depicting a synoptic overview (now Fig. 1).

**Response to general comments:**

- The science questions are good, but radiative effects that were a big part of the motivation for studying WCB microphysical properties aren't examined. Is there a reason for that? Connecting microphysical changes to radiative changes would be very insightful.

UTLS moisture is important for Earth's climate and we find large differences in the ice content and relative humidity in our study, which indicates that these results could also have an impact on the radiative budget of WCBs. Sadly, the original study by Oertel et al. (2025) did not focus on radiative effects and therefore did not write relevant variables to file. Nor are the simulations long enough to do a robust analysis of radiative impact. Therefore, we are not able to investigate what the impact of our findings are for the radiative effect of WCBs. However, it is important to point out that this could be a topic for future research, given the large differences we found in UTLS moisture conditions between PPE members. We therefore add an item to the final outlook bullet-point list following the point discussing the implications for geoengineering:

*"Following this, future studies could investigate how the different UTLS moisture conditions produced by the PPE members influence the radiative effect of WCBs."* (line 771)

- A few questions on sensitivities to the methodology:
There is a bit of justification provided for the range of some parameter values, but it isn't clear how all of the CAP, INP, CCN, SAT, and SST ranges are chosen for the PPE. Presumably these are constrained by possible real world values dictated by previous studies?

This is an important point and we add some additional information in the paper, based on the PPE design by Oertel et al. (2025), to make this more clear (Line 236).

- For the CCN scaling factor, values were chosen such that they approximately represent observed CCN variability ranges in the North Atlantic regions. We have added this information to the end of the paragraph on the CCN scaling factor:

  *"This variation results in CCN concentrations varying from approximately 100 $cm^{-3}$ to 5,000 $cm^{-3}$, which approximately represents the variability that has been observed for CCN concentrations (Hande et al., 2015, 2016; Genz et al., 2020; Wang et al., 2021)."*

- For the INP scaling factor, we now add the information that "*their concentrations in the atmosphere vary over several orders of magnitude*" (Line 205). Later in the paragraph, it is already noted by how much the concentrations approximately vary given the perturbations.

- For CAP, we already provide this information and cite relevant literature (Line 197):

   "*While theory predicts a normalized CAP value of 0.5 for perfectly spherical particles, realistic ice and snow hydrometeors often exhibit considerably different values (Westbrook et al., 2008; Chiruta and Wang, 2003). Consequently, in the PPE, CAP for ice and snow is varied (simultaneously) by a scaling factoring ranging from 0.2 to 1, which explores a range for CAP from 0.1 to 0.5.*"

- For SAT we add the following information (Line 249):

   "*For a factor f-SAT AD of 0.1, vertical velocities of 2 ms$^{-1}$ can produce supersaturations of 20%, which aligns with theoretical examinations of supersaturation in liquid clouds (Korolev and Mazin, 2003; Morrison and Grabowski, 2008)*"

- For SST we believe that the paragraph is already sufficient to explain the +-2K range for variation.

With no test dataset for the RF model, couldn't that result in overfitting to this single case?

That is correct. However, this is not an issue in this case because of how we use and interpret the RF models. We use the RF models purely as a diagnostic tool—to rank parameter sensitivities and explore partial-dependence curves—not to make out-of-sample forecasts. If a model has a good fit — that is, if the root mean square error (NRMSE) is low — then it has determined which parameters and thresholds are important for "predicting" a target variable. We can then use these features to gain insight into what the model learned.
Even so, we guard against over-fitting by making sure the RF models did not have the capacity to memorize the data (number of trees, depth, etc.). Thus, the NRMSE is never small enough to indicate overfitting.

WCBs need some definition so thresholds are to be expected, but are results sensitive to the 600-hPa ascent depth and ascent rate threshold used?

The criterion that an air parcel must ascend at least 600 hPa in no more than 48 hours to be considered part of a WBC is a widely accepted threshold adopted in nearly every WBC study. A recent study by Heitmann et al. (2024) determined that relaxing this criterion results in only minor changes for various analyses (https://doi.org/10.5194/wcd-5-537-2024). Therefore, we are confident that our results are not substantially affected by this threshold.

For sensitivity to CCN, how much should we trust the snow and graupel changes in terms of applicability to the real world given arbitrary threshold conversions between these 2 categories as opposed to riming transitioning smoothly to produce a range of variably rimed precipitating ice?

This is a good point that was also pointed out be RC1 in a similar way. However, since this is a modeling study, we cannot make any definite statement on the effects in the real world without additional observational studies. Nevertheless, we have modified this paragraph to reflect that this process is continuous in reality (Lines 465-470) :
"*The mean of the maximum mixing ratio for snow (max(qs)) during ascent also increases with CCN, while for graupel (max(qg )) it decreases (Fig. 6 e). We explain this as follows: increased CCN concentrations lead to more numerous but smaller cloud droplets, which slows the formation of raindrops. This, in turn, reduces the collision efficiency between ice or snow particles and cloud droplets, slowing the (continuous) conversion to graupel. As a result, snow remains more abundant and graupel less abundant. However, this large effect of the CCN scaling factor*

*on the hydrometeor population during the ascent is mostly lost by the end of the ascent, showing that once air parcels glaciate, CAP and INP dominate the ice-phase cloud microphysics"*

Figure 6 also shows nonlinear relationships where CAP, INP, and CCN sensitivities are particularly low or high, so it should perhaps be noted that the distribution of values of these parameters in the real world is important for dictating whether overall sensitivities are large or small.

This is a great point which we now address in Line 494:
*"Notably, these relationships seem non-linear when CAP, INP scaling and CCN scaling are particularly low or high, indicating that the distribution of these parameter values in the real-world are important for determining whether the overall sensitivity of RHi to perturbations is large or small."*

With the greater sensitivities in fast ascending trajectories, it seems like model representation of convective processes could be important and there could be some model resolution dependence there. Should that be mentioned?

For fast-ascending trajectories, only the sensitivities for the ice mixing ratio Ni and the ice radius ri are larger. The sensitivity for relative humidity over ice is smaller. We now point this out in Line 616:
*"We also note that for fast ascending trajectories, Ni and ri are more sensitive to CAP and the INP scaling factor than for all trajectories, suggesting that overall sensitivities to these parameters might increase for higher resolution simulations, where a greater fraction of trajectories ascend quickly."*
And in Line 720:
*"Ni and ri are more sensitive to CAP and the INP scaling factor for fast ascending trajectories than for all trajectories. This indicates that simulation scale might also influence Ni and ri, since Choudhary and Voigt (2022) found that a greater fraction of WCB trajectories ascend quickly when model resolution is increased."*

The conclusions and discussion should include caveats. For example:
- This is a single case, and it isn't clear how representative it is of WCB events in general.
- PPEs sample the uncertain multi-parameter phase space but still have the weakness of assuming constant parameter values for some parameters that are not real world physical constants. Thus, sensitivities can be overestimated relative to a potentially more realistic stochastic framework in which constant parameters may be varying (e.g., Stanford et al. 2019).
- With only Hallett-Mossop rime splintering parameterized for secondary ice production, could mixed phase ice concentrations be biased low, potentially influencing the WCB outflow sensitivities? Recent studies by Alexei Korolev, Vaughan Phillips, and others have highlighted the potential importance of additional secondary ice mechanisms such as raindrop fragmentation upon freezing and ice collisional breakup.

We have included these (and other) caveats in Line 750: *"However, this study compromises of only one case study, and it isn't clear how representative our findings are of WCBs (or other model configurations and microphysics schemes) in general. Additionally, PPEs sample the uncertain multi-parameter phase space but assume constant parameter values throughout the simulation for some parameters that are not constant in reality. Therefore, sensitivities can be overestimated relative to a potentially more realistic stochastic framework in which parameters vary during the simulation (e.g Stanford et al. (2019)). Furthermore, the microphysics scheme employed in the ICON simulation only considers rime splintering; it does not account for raindrop fragmentation upon freezing or ice collisional breakup for secondary ice production. This could result in biases in mixed-phase ice concentrations. Finally, there are likely additional important parameters that influence the variables examined in this study, but that were not perturbed (see Hieronymus et al. (2025))."*

- Could results be sensitive to the thresholds used to define the WCB (600-hPa depth) and their ascent rates?

Given our answer to this question above, we have not included this caveat. The ascent rates are discussed in the text.

Sentences beginning with "This" could be made clearer by stating the object that it is referring to (for example, "this difference…" instead of just "this…"). These are several instances:

Thank you for these suggestions which make the text more readable. We have implemented these changes and looked for other instances where we write "[…]. This is / makes / means / etc. […]" and have rewritten for clarity whenever it enhances readability.

Lines 67-68: How far south are the authors referring to? Could the authors be more specific?

We have rephrased to (Line 80): "[…] *when WCBs are usually located closer to the equator than during later stages, […]*". Also, we have added a synoptic overview of the case in the new Figure 1.

Lines 267-271: Are the authors saying that a high IBF score could be due to a parameter being highly correlated with another parameter instead of the parameter being "actually important"?

No, we mean that a high IBF score without a clear correlation with the target variable means that the variable might be important for the decision making structure of the RF model in *combination* with another parameter (PPE parameters do not correlate with each other, they are sampled according to the latin hypercube method). We try to clarify what we mean by rephrasing this sentence as follows (Line 295):
"*Note: a high IBF importance score for a PPE-parameter does not necessarily imply a clear or strong correlation with the output variable. Instead, it indicates that the PPE-parameter contributes strongly to the RF-model's predictions, possibly by affecting the RF-model's decision-making structure only in combination with other PPE-parameters, and is only a meaningful metric when the RF-prediction is good. "*

*Lines 304-305: Since the authors are discussing temperatures in terms of Celsius, could the corresponding plots (e.g., Fig. 2) be modified to be in terms of Celsius? Using Celsius would make more intuitive sense in the framework of microphysics discussions including homogeneous freezing.*

We modified all temperatures in the figures and in the text to degrees Celsius.

Line 325: For the first part of this sentence before the comma referring to all PPE members having a mean RHi > 100%, could the authors refer to Figure 2j?

Yes, we have added the reference to the figure.

Lines 325 to 327: What about the observation made is "particularly interesting"?

We find it interesting that changes to RHi are large eventhough changes in qv are small. RHi is largely dependent on vapor content, so if RHi changes by a lot, it is interesting that qv does not. We state this more clearly now (Line: 355) "*[…] which is particularly interesting **because** the differences in qv are small.*"

Lines 341 to 345: It seems like the argument here is that using the means for the RF model means that the spread of the distribution (5th to 95th percentiles) is not considered by the RF model. If so, the argument as stated appears a little convoluted and difficult to follow. Is what matters here the change in means between PPE members relative to the spread between the 5th to 95th percentiles (because all variables could be argued to have a large 5th to 95th percentile spread)?

In this paragraph, we aimed to point out that any changes in the mean or median of p or T should not be taken too seriously given the large spread of p and T values, which is much larger than any change in the mean or median of p or T. Furthermore, we state that the RF model does not identify meaningful relationships between PPE-parameter perturbations and these values anyway. We realize that this paragraph is somewhat redundant and have modified it to clarify this point (Line 362):

*"As discussed above, T , p and qv show little variation between PPE members (Figure 3 b, d and f), with the variability within each PPE member being significantly larger than the mean differences between PPE members. The distributions for PPE members with the highest and smallest mean values for T , p and qv are also very similar (Figs. 3 a, c and e). The relatively large spread of the distribution (min/max shaded areas in Fig.3 a, c, and e) is a result of the differing number of WCB trajectories per PPE member (which is primarily controlled by SST, see Oertel et al. (2025)). The RF2*

*mean(p) and RF2 mean(T) model predictions (which we use to determine which parameters have the strongest influence on a variable, as long as the model prediction is strong) are also relatively weak, with mediocre R2 and large NRMSE-values (Tab. 2). Therefore, the RF model does not find meaningful changes in T_mean, p_mean and qv_mean with parameter perturbations. We interpret this inability of the parameters to change the pressure, temperature, and vapor conditions at the end of the WCB ascent as an indication that the thermodynamic conditions in the outflow of a WCB are largely constrained."*

Lines 347: It is unclear how Figure 2c shows a correlation between T95 and qv95. Perhaps the authors meant to refer to Fig. A1c?

Yes, thank you.

Line 354: This statement ("The change is stronger for qv_95 than for T_95") presumably refers to Figs. 2f and 2b. It is unclear how this change is computed and how the 2 different variable changes can be fairly compared against each other. Perhaps the max 95th value minus the min 95th value divided by the 5th to 95th percentile spread to compare changes relative to the range of variable values?

We have removed this sentence because upon examining the figures and data again, we find that qv_95 does in fact NOT change more strongly with SST than T_95.

Line 359: It is clear visually that the highest and lowest qv value correlates strongly with the calculated saturation specific humidity. However, could the authors include a correlation coefficient to quantitatively support this claim?

We have added information on the spearman correlation coefficient of qv vs qv_sat for both groups (0.99) in the text.

Lines 357 to 361: Isn't Fig. A1a or something similar to it plotting qv as a function of temperature a more straightforward argument than Fig. A1b (qv vs. qv_sat) that qv is strongly constrained by temperature? Qv correlates strongly with qv_sat, but they are not 1:1, and it isn't clear from Fig. A1b alone how temperature vs. pressure modulate qv_sat to affect that relationship or how dynamics and microphysics affecting supersaturation.

We want to point out that qv at the end of the ascent is primarily constrained by the thermodynamic conditions at the end of the ascent for all simulations, and this includes the pressure as well as the temperature. Plotting qv vs temperature would only show how strongly qv correlates with temperature; using qv_sat instead also enocdes the dependence on pressure. Hence, we plot qv vs qv_sat for two simulations that have the most different 95[th] percentiles for qv, to show that for both simulations, qv at the end of the ascent correlates with qv_sat (and is therefore primarily constrained by the thermodynamic conditions).

Lines 376: "The spread… is unchanged…" The word unchanged seems a little too strong. Could the authors moderate it to "**mostly** unchanged"?

Yes.

Lines 377 to 378: "… reduces the spread". This reduction is not easily visible. Could the authors quantify this reduction?

We have added information on the spread of values to back up this claim (Line 404).

Line 380: The second mode in the distribution is not a "peak" since it is not a local maximum. It would be more accurate to describe this as a "second mode."

This is a more accurate description and we have made sure to call this second peak a "mode" throughout the text

Line 390: "…many small cloud droplets reach the homogeneous freezing level." How is "many" defined here? It is highly likely this is homogeneous freezing and glaciation temperatures in Fig. S11 provide some support, but could it be shown that this second mode is indeed due to homogeneous freezing, e.g., by examining drop concentrations at -38C or the change in ice concentration across that temperature level?

We meant to say the following, which we now write instead (Line 419):
*"More homogeneously freezing cloud droplets when CAP is low explains why the distribution appears slightly bi-modal at low CAP values — indicating **that a large proportion of trajectories contain small cloud droplets that reach the homogeneous-freezing level** — but becomes uni-modal at high CAP values."*

Concerning the number of cloud droplets reaching the homogeneous freezing level: due to vertical interpolation errors and time resolution of the trajectory data, it is not possible to examine drop concentrations just below -38°C accurately. Nevertheless, we are confident in our interpretation of the data, because the only physical process that can produce ice-crystal number mixing ratios in the order of $10^7 - 10^8$ kg$^{-1}$ is homogeneous cloud freezing. Heterogeneously produced ice-crystal number mixing ratios cannot exceed the number of INPs, which is on the order of $10^4$ kg$^{-1}$.

Figure S12 caption: "second peak" should be "second higher concentration mode."

We have implemented this change

Lines 423 to 424: "This clear dependency… parcel evolution (Fig. 5e)": In Figure 5e, the max_qc panel is dark red, and the max_qr panel is dark blue. Shouldn't these large magnitudes mean that CCN strongly modifies the liquid

content rather than not strongly affecting it? Also, is "in early stages of parcel evolution" inferred from the liquid mass mixing ratios being maximum values?

Thank you very much for pointing this out; we are not sure why this false statement was made. CCN definitely effects the liquid water content during the ascent and we have modified the sentence to reflect this.
Regarding the second question: yes, the early stages are inferred from max_qc and max_qr being achieved during the ascent.

Line 424: "Therefore, cloud droplets are far smaller…" It is unclear how cloud droplets are proven to be smaller when qc strongly increases with the CCN scaling factor in Figure 5. Is there other evidence to support this assertion?

We have done some additional analysis reformulated these sentences to now write (Line 457):
*"With an increase in CCN, cloud droplets during the ascent become smaller; for simulations with the largest and smallest CCN scaling factor, the mean cloud droplet mass is approximately $72 \cdot 10^{-9}$g and $9 \cdot 10^{-9}$g, respectively. Smaller*
*cloud droplets delay the formation of rain and enhance the residence time of liquid condensate. This delay […]"*

Line 428 to 431: the word "presumably" is used twice in this sentence. Suggest replacing one of them with a synonym for improved readability.

We have changed this entire paragraph due to a more physically sound explanation suggested by RC1. We have summarized it about, in the question about the continuity in conversions from ice to graupel.

Lines 458 to 459: How robust are these red and cyan best fit lines? Could the authors include correlation values?

 We have added spearman correlation coefficients for the red and cyan lines in the text.

Line 462: "… the spread of RHi values reduces strongly…" Is there a way to quantify this spread and its reduction?

We have added information in the text on the standard deviations of RHi values for simulations that have the maximum and minimum values for CAP, INP and CCN scaling.

Line 473: "The mean and median for the first group…" Could the authors clarify what "first group" and "second group" are referring to?

We have added clarification by adding "(CAP and INP scaling large)" and "(CAP and INP scaling small)" behind the respective groups

Line 501: Referencing specific visual cues would greatly help the readability of the sentence: "**as indicated by the red line**, CAP reduces…"
Line 509: "… below this threshold for the second group…" Could the authors clarify which line in the figure they are referring to?
Line 510: "…higher abundance of trajectories above the threshold also explains the long tail". Could the authors clarify that they are referring to the lines in Fig. 8A?

We have modified this paragraph (Lines 445 to 455) to make all references to lines in plots, threshold values and trajectory "groups" more clear.

Line 553: The authors mention fast vs all trajectories, but the figures the authors are referring to within this sentence only contain fast trajectories. Could the authors reference back to the appropriate figure for comparison between the two trajectory groups? For example: "…for fast ascending trajectories than for all trajectories (compare xxx in Figures X and X)".

We reference appropriate figures and make more clear when we refer to differences in fast trajectories and all trajectories.

Lines 557 to 560: "This is because the peak t for fast trajectories **in Fig. 12b** is below 5*10^2…"

We have implemented this

Line 586: "The large Ni implies smaller Tsat…" Recommend referencing equation 2 here. Also, since decreasing ri has the opposite impact on Tsat, the authors should mention why the effect of Ni on Tsat is greater than ri.

We have implemented this

Line 605: Can the authors describe which behavior they are referring to?

We have modified the sentence (Line 658): "*which is probably a reflection of the **larger Ni** found at the end of the ascent*"

Line 606: Could Fig. 4e be referenced right after "bimodal distribution" for improved clarity?

We added the explanation that this is because of the increased number of high Ni values when CAP is small (while implementing this change we realized that we had falsely written "when CAP is large" and have changed this to "when CAP is small").

Line 643: "… a small shift in the lower Ni, which…". It appears that there is a word missing after "Ni".

We have modified the sentence to make it more understandable.

Line 697: Did the authors mean "Ni 5 hours after ascent"?

Yes, thank you

Throughout: When using terms like mean maximum variable, it would help to clarify what the mean and maximum correspond to, so it is clear how the variable is being computed.

We have changed these phrases to be phrased like: "*mean **of the** maximum mixing ratio […]*"

(Optional) The authors should consider re-spelling out the acronyms at key locations (e.g., figures, conclusion) to aid readers.

Figure 2: Make sure that to clearly state that largest and smallest "means" refer to the distributions of the variable for ensemble members with the smallest and largest mean value of that variable. The caption is a bit confusing as currently worded.

Figure 3: Could the authors mention in the caption that these distributions correspond to those at the "end of ascent"?

Figure 5e: (optional) Could the correlations be shown in a table format to add additional information such as the correlation coefficient value? Currently, max_qc and max_qs are very similar in shading.

Thank you for these suggestions. We have implemented most of them. The last comment we chose not to implement because the exact correlation coefficient is not important, only that the correlation is strong.

---

## Author Comment (AC4)

**Response to RC1**

We would like to thank the reviewer for their thorough reading of the paper, their valuable comments, and their recommendation to accept it for publication. The reviewer raised important points and made smart observations that improved the paper. When we do not implement one of their recommendations, we provide a clear explanation of our reasoning.

Below, we first respond to their major comments, then to their minor comments. We have implemented the technical corrections and omitted them from the response.

More changes than listed have been implemented to accommodate the suggestions by the other three reviewers. Also, figure numbers are shifted by one due to the incorporation of a new figure depicting a synoptic overview (now Fig. 1).

**Response to major comments:**

*– In Oertel et al. (2025), and in this paper, a multidimensional Gaussian Process (GP) emulator is used to approximate model behavior. This approach assumes that input parameters, such as capacitance, vary smoothly and that the resulting model output — including variables like ice number concentration and mean ice particle size — responds in a continuous and predictable way.*
*Although internal nonlinearities exist in the Seifert and Beheng (SB, 2006) microphysics scheme (e.g., mass–size relationships, category transitions), these are embedded within the model physics and do not necessarily introduce discontinuities in the macroscopic output variables if each hydrometeor category is analyzed separately.*
*However, imposed minimum or maximum limits on ice number concentration may lead to plateaus or discontinuities in mean ice size, particularly when ice mass continues to grow while the number concentration is constrained. Additionally, ICON's saturation adjustment scheme limits supersaturation to 100% in regions containing cloud droplets or rain, potentially producing flat regions in the relative humidity w.r.t. water output fields. Given that a Gaussian Process emulator is applied, do these plateaus or discontinuities introduce challenges (if they exist in any of the output fields) for the emulator's ability to accurately capture the underlying parameter–response relationships?*

While Oertel et al. (2025) use a Gaussian process emulator to approximate model behavior, we do **not** use one in this paper. We strictly analyze the model output from the 70 PPE members without additional processing. The results obtained from the GP emulator analysis in Oertel et al. (2025) are **not** used, discussed, or examined in this paper. Therefore, this question does not apply to this study. If the reviewer is instead referring to the use of random forest (RF) models, we point out that we do not use these to make predictions or interpolations, but only to identify important parameters and their interactions. Also, there is no indication that there are any plateaus or discontinuities in the data.

*– In these ascending parcels considered within the WCB, the updraft speed is not zero. For example, when $w = 0$ and $Ni_{ri} = 10^{-1}\ \mu m\ cm^{-3}$, the relaxation time can be as large as $10^4$ s (Fig. 1). If $w = 1$ m/s, the relaxation time can be reduced by approximately an order of magnitude under upper tropospheric conditions. For larger values of $Ni_{ri} > 10^1\ \mu m\ cm^{-3}$, the impact of $w \leq 1$ m/s becomes negligible (e.g., Korolev and Mazin (2003)).*
*Assuming $w = 0$ for all analyses of ascending parcels may lead to severely overestimated the ice phase relaxation times, particularly for fast-ascending trajectories.*
*It is stated in Line 492 that $w \approx 0$ for parcels taken at the end of ascent, but later, for example in Section 4, fast ascending trajectories are analyzed together with relaxation timescales. It is therefore recommended to explicitly include $w$ in the relaxation timescale formulation to make this analysis more robust*

The reviewer raises an important point, but we argue that no changes are required.

First, in this paper, we examine the relaxation timescales of trajectories at the "end of the ascent," which, for each trajectory, is the point at which it has **completed** an ascent of 600 hPa and is **no longer rising faster than 8 hPa/h**. In the hour after this point, the mean vertical velocity for all trajectories is -0.0084 m/s, with a standard deviation of 0.046 m/s (see left figure below). For the fast ascending trajectories the vertical velocities in the hour after the end of the ascent are on average -0.0048 m/s, with a standard deviation of 0.054 m/s (right figure below). They may have high vertical velocities **during** the ascent but are no longer ascending when examined. Therefore, modifications to the relaxation timescale to incorporate the vertical velocity will be minimal.

However, even if neglecting vertical velocity would lead to biases, we point out that we do not use the relaxation timescale for any calculations or further analysis, but only as a **diagnostic** to explain the differences observed in relative humidity. The correlation between the saturation timescale and relative humidity is clearly shown in the paper. Therefore, explaining RHi using the relaxation timescale is plausible, regardless of a potential bias.

Therefore, we see no need to include vertical velocity in the relaxation timescale formulation. We hope the reviewer now shares this opinion.

[Figure]

[Figure]

*– Line 430: "The mean maximum mixing ratio for snow (max(qs)) during ascent also increases with CCN (Fig. 5 e), presumably because in the two-moment microphysics scheme riming contributes to the mass growth rate of snow,...". The reasoning here may not agree with how riming affects particles in SB. In this case: When riming occurs it can have a two fold effect on ice and snow. If ice or snow rimes with raindrops it is converted to graupel else if it rimes with cloud droplets it remains either ice or snow unless it exceeds a critical rime mass threshold depending on some tuning parameter. Typically snow is considered as pristine and therefore is converted quite rapidly to graupel (by assuming a low space filling constant). If riming on snow occured more often then qs would most likely decrease because of this conversion, not increase, right?*
*I would suggest the process look more like the following: Increased CCN concentrations lead to more numerous but smaller cloud droplets, which slow the formation of raindrops. This, in turn, reduces the collision efficiency between ice or snow particles and cloud droplets, slowing the conversion to graupel. As a result, snow remains more abundant.*

We thank the reviewer for this idea, which is indeed a much better explanation than ours. We have adopted this and now write the following (Line 465): "**The mean of the maximum mixing ratio for snow (max(qs)) during ascent also increases with CCN, while for graupel (max(qg)) it decreases (Fig. 6 e). We explain this as follows: increased CCN**

**concentrations lead to more numerous but smaller cloud droplets, which slows the formation of raindrops. This, in turn, reduces the collision efficiency between ice or snow particles and cloud droplets, slowing the conversion to graupel. As a result, snow remains more abundant and graupel less abundant.**"

*– Line 394: I follow this reasoning, but a similar argument applies to lower INP scaling: with fewer activated INPs, there is less competition for available vapor, resulting in larger mean ri. This in turn enhances the conversion from qi to qs. Some aspects of this reasoning might become clearer, and confirming the authors explanation, when qs is also plotted in reference to Fig. 4d, e, and f*

This is a smart observation. We have checked our data and found that when INP is small, the snow content at the end of the ascent does slightly increase, as hypothesized by the reviewer.

However, the amount of snow at the end of the ascent is very small (qs is on average $10^{-3}$ g/kg and Ns on average 500 1/kg, compared to $5*10^{-2}$ g/kg for qi and $2.5*10^6$ 1/kg for Ni). This is because of the large abundance of ice crystals and because most snow particles precipitate by the end of the ascent. Therefore, the effects of the INP scaling factor on the ice content at the end of the ascent are still dominated by the effects we describe (the sheer amount of ice crystals drown out the process described by the reviewer).

However, the process proposed by the reviewer does explain the tail to larger ri for small INP. We have therefore added the following statement at the end of the paragraph titled "Impact of INP on ice properties at end of ascent" (Line 448): "**The tail towards larger ri for small INP can be explained by the fact that when fewer ice crystals form during the ascent, there is less competition for available vapor, allowing for individual ice crystals to grow larger. This also enhances the conversion from ice to snow, which is supported by the fact that on average, qs at the end of the ascent slightly increases when INP is small (not shown).**"

We have not added another Figure showing the effect on snow, because the snow content at the end of the ascent is negligibly small, and we think that this would broaden the scope of the Paper too far.

**Response to minor corrections (excluding technical corrections):**

*– Line 27: Instrument uncertainties remain, with calibrated measurements typically accurate to within 5–10% in supersaturated regimes (Petzold et al., 2020). Can it be added to highlight uncertainty with instrument*

We have added this information on instrument uncertainties, citing Petzold et al. 2020 in line 28.

*– Line 42: "tropopause region has increased on average from 2011 to 2020 compared to the 1980s". Can a more specific value be added?*

We have contacted the authors and revised this statement to more accurately reflect their findings. They do not focus on changes in the transport of substances but instead on changes in the vertical transport of air-masses. Therefore, the revised sentence now reads (Line 42): "**[…] but a historical climatology by Jeske and Tost (2025) found that the height of convective outflow has shifted to lower pressures from 2011 to 2020 compared to the 1980s.**" A more precise and quantitative statement is not possible given the findings from their paper, which conducts a largely qualitative analysis.

*Line 48: "key contributor to extratropical UTLS moisture". By how much do WCBs contribute?*

The measurements by Zahn et al. 2014 did not enable a quantitative analysis of how much WCBs contribute to UTLS moisture, only the qualitative conclusion that they play a large role overall. However, ongoing climatological research by Ziyan Guo at the University of Mainz has found that in December, January and February, the WCB moisture transport by grid-scale advection accounts for up to 13.8% (23.3%) of the total water (condensate) transport into the upper troposphere

above the North Atlantic and Pacific ocean basins. At this time, the publication has not yet reached a stage where we can cite it. **We have therefore, regrettably, not changed this line (yet). If the publication in question is published during the typesetting stage, we will add this information and citation here.**

*– Line 59: "various atmospheric constituents". Be more specific and replace the phrase with ...water vapor, hydrometeors and aerosols..*

The sentence now reads (Line 68): "**In addition to producing precipitation, WCBs transport considerable amounts of energy as well as water vapor, hydrometeors, aerosols and trace gases across latitudes, and influence large-scale weather patterns […]**"

*– Line 64: What is meant by the "incorrect representation of WBCs"? What is misrepresented? The location, vertical extent, cloud dynamics, cloud microphysics or water vapor transport?*

Thank you for pointing out this sentence that could be more precise. We have changed it to better reflect what we want to convey and it now reads (Line 74): "**Given that WCBs exert such wide-ranging influences, forecast skill is highly sensitive to how well models capture their path, vertical extent, diabatic-heating profile and mixed-phase cloud evolution, and any uncertainty in representing these aspects can quickly degrade predictions of cyclone intensity, heavy-precipitation placement, downstream wave development, and even heat waves (Pickl et al. [...]**"

*– Line 70: Cloud overlap can also change a cooling effect into a warming effect during the day (Johansson et al., 2019)*

We have added this information. The sentence now reads (Line 81): "**As the WCB progresses northward, high-level frozen clouds have a warming or cooling effect depending on solar insolation, cirrus optical thickness, and potentially cloud overlap (Krämer et al., 2020; Joos, 2019, Johansson et al., 2019).**"

*– Line 73: Is the warming effect the average when considering day (cooling) and night (warming)?*

Yes. We now state (Line 85): "**… that has an average net warming effect.**"

*– Line 105: "Supersaturation" is a continuous variable. Change the sentence to: "otherwise, greater or more widespread supersaturation would be produced."*

This is a good suggestion that we have implemented as is suggested (Line 118).

*– Line 118: If the cloud is glaciated then CCN becomes less relevant because of the Wegener-Findheisen-Process. In this case would CCN matter? Before glaciation CCN may be very important*

We have added clarity to this statement by modifying the sentence that follows immediately afterwards, which now reads (Line 129): "**For instance, an increase in the capacitance could accelerate ice particle growth, while higher concentrations of CCN can lead to more and smaller cloud droplets in the liquid phase, which can in turn increase the number of ice crystals in the frozen phase and enhance vapor conversion.**"

*– Line 165: I realize that is given in previous papers, but it would be good to have a very short one paragraph overview here with Figure 1 from Oertel et al. (2025).*

After consideration we agree that this will help readers better conceptualize the case study without having to consider additional literature. We therefore have added a short paragraph as well as an additional plot (now Figure 1) that shows the synoptic evolution during the case study period using ERA5 data.

*– Line 219: Replace the sentence with "In the PPE, the scheme is modified to allow supersaturation to develop under*

*conditions of strong vertical velocity."*

This is a good suggestion that we have implemented as is suggested (Line 243).

*– Figure 3 caption: Keep the caption between Fig 3 and 4 consistent... is it median qi, median Ni median ri or only qi, Ni, ri? I believe it should Fig. 4's description is incorrect.*

We have made sure that the figure captions are correct and consistent (now Figures 4 and 5).

*– Line 444: I don't fully understand this hypothesis. Could you please provide one or two additional clarifying sentences? Does it suggest that stronger convection over higher SSTs would lead to increased Ni and qi? If so, it seems that the primary driver would not be the SST itself, but rather the associated increase in vertical motion*

Yes, we believe that if the WCB had more convection, we would also see a greater change in convective activity for simulations with high and low SST, which would would then influence Ni and qi more strongly than in our WCB case. **We have rewritten this entire paragraph (Lines 471 – 485) and hope that we make this point more clear.**

*– Line 551: It is a interesting idea. Can you plot the cloud droplets numbers concentration in each case just before the homogeneous freezing temperature? E.g. are there more cloud droplets available for CAP < 0.4 and INP scaling < 0.5? If so, this will solidify your reasoning*

Because the physical processes in the model are represented on discrete vertical levels, and because the vertical interpolation required to transition from an Eulerian to Lagrangian perspective when analysing cloud droplet number mixing ratios just before reaching the homogeneous freezing temperature is not straightforward, we refrain from showing this analysis. However, we can solidify our reasoning as follows: the only physical process that can produce ice-crystal number mixing ratios in the order of $10^7 - 10^8$ kg$^{-1}$ is homogeneous cloud freezing. Heterogeneously produced ice-crystal number mixing ratios cannot exceed the number of INPs, which is on the order of $10^4$ kg$^{-1}$. We have added this information to support our conclusions on homogeneous freezing the first time we talk about the process in Line 417:
**"This hypothesis is supported by the fact that the only physical process producing ice-crystal number mixing ratios in the order of $10^7 - 10^8$ kg$^{-1}$ is homogeneous cloud freezing. Heterogeneously produced ice-crystal number mixing ratios cannot exceed the number of INPs, which is on the order of $10^4$ kg$^{-1}$."**

*– Line 508: If one further assumes that the parcel is subsaturated, or that collisional growth is negligible and no conversion to snow occurs under supersaturated conditions, right?*

That is correct and should be addressed. We have expanded on this sentence, which now reads (Line 662): **"However, the absence of higher median Ni 5 h after the ascent suggests that these small ice crystals sublimate. The alternative explanation to sublimation is collisional growth, which is proportional to the particle number concentration ($\sim 10^5$ kg$^{-1}$),which is much smaller than in a typical mixed-phase cloud ($\sim 10^8$ kg$^{-1}$), and the sticking efficiency, which is temperature dependent and very small at these low temperatures (Seifert and Beheng, 2006, 2001). Therefore, we find sublimation the most likely explanation, given the large fraction of trajectories that are sub-saturated with respect to ice during this time (Fig. 16 c)."**

*– Line 692: How would one constrain realistic values for CAP if it is a function of the habit of ice crystals which keeps changing depending on the state of the environment?*

That is a good point. We modify this statement to not suggest that more *realistic* values could be identified, but instead more appropriate values, which might not be realistic by themselves, but would produce more realistic *results*. The sentence now reads (Line 763): **"[…] may help in selecting appropriate values for CAP and the INP scaling factor for WCB flow configurations."**.

*– Line 704: In the introduction Earth's Radiation budget is mentioned and the sensitivity to vapor content in the UTLS. How does the results presented here have an impact radiation*

Sadly, the original study by Oertel et al. (2025) did not focus on radiative effects and therefore did not write relevant variables to file. Therefore, we are not able to investigate what the impact of our findings are for the radiative effect of WCBs. Nevertheless, it is important to point out that this could be a topic for future research, given the large differences we found in UTLS moisture conditions between PPE members. We therefore add an item to the final outlook bullet-point list following the point discussing the implications for geoengineering (Line 771): "**Following this, future studies could investigate how the different UTLS moisture conditions produced by the PPE members influence the radiative effect of WCBs.**"

---

## Referee Report (RR1)

Overall, I am satisfied with how the authors addressed my questions and comments during the first round of reviews. I recommend acceptance. My specific suggestions for improvement are outlined below.

**Minor Comments**

- Line 27–28: "...with calibrated measurements typically only accurate to within 5–10% in supersaturated regimes..."

- Line 85: "...has an average net warming effect" is redundant. Suggested: 'has a net warming effect on average."

- Line 87: The use of water vapor and moisture is inconsistent. Consider using only one of the two throughout where applicable.

- Line 98: The phrase "...and hence the parameterization of the microphysical processes within them are a major source of uncertainty" contains a subject-verb agreement issue. Suggested: "...and hence the parameterizations of the microphysical processes within them are a major source of uncertainty."

- Line 105: "This has lead to the employment..." contains a grammatical error. Corrected: "This has led to the employment..."

- Line 209–210: "...by a logarithmically scaled factor ranging from 0.01 to 20." Suggested: "...by applying a logarithmic scaling factor between 0.01 and 20 to emulate varying INP concentrations."

- Line 235–236: "...by scaling the number of activated cloud droplets per time step in the entire profile..." Suggested: "by scaling the number of activated cloud droplets throughout the vertical profile at each model time step..."

- Line 275: "The values of the variables are usually taken..."

- Line 342: "droplet mass concentrations" should be droplet mass mixing ratio

- Line 351: Inconsistent use of units. Replace g/kg with $kg\tilde{k}g^{-1}$ for consistency.

- Line 396: "concentration" should be "mixing ratio"

- Line 397: "frozen hydrometeors" or "frozen phase" is typically referred to as "ice hydrometeors" or "ice-phase" Apply this terminology consistently throughout.

- Line 462: Should the definition be : (qc + qr)/qtot ?

– Line 503–504: "...slightly increase the median $RH_i$..." Corrected: "...slightly increases the median $RH_i$..."

– Line 534: ' "proportional to the ice number concentration times the radius of ice..." Suggested: "inversely proportional to the product of ice number concentration and ice particle radius..."

– Line 589: "$\sim$ –43 °C and $\sim$ –330 hPa". Typo in pressure: "$\sim$ –43 °C and $\sim$ 330 hPa"

– Line 658: Keep consistency with hour units. Sometimes hour is used and other times h.

– Line 682: "...modulated by cloud physics processes". Slightly awkward. Consider: "modulated by cloud microphysical processes".

– Line 703: "to a lesser extend". Typo. Suggested: "to a lesser extent"

– Line 714–715: "with the a less pronounced impact". Typo. Suggested: "with a less pronounced impact"